# Analysis of the spatial association of geographical detector-based landslides and environmental factors in the southeastern Tibetan Plateau, China

Wei-Jie Jia[1,2,3]*, Meng-Fei Wang[3], Cheng-Hu Zhou[1], Qing-Hua Yang[3]

**1** State Key laboratory of Resource and Environmental Information System, Institute of Geographic Sciences and Natural Resources Research, China Academy of Sciences, Beijing, China, **2** University of the Chinese Academy of Sciences, Beijing, China, **3** Department of Geological Remote Sensing, China Aero Geophysical Survey and Remote Sensing Center for Natural Resources, Beijing, China

\* sunleaver@gmail.com

**Data Availability Statement:** All relevant data are within the paper.

**Funding:** This research is funded by the China High-resolution Earth Observation System

## Abstract

Steep canyons surrounded by high mountains resulting from large-scale landslides characterize the study area located in the southeastern part of the Tibetan Plateau. A total of 1766 large landslides were identified based on integrated remote sensing interpretations utilizing multisource satellite images and topographic data that were dominated by 3 major regional categories, namely, rockslides, rock falls, and flow-like landslides. The geographical detector method was applied to quantitatively unveil the spatial association between the landslides and 12 environmental factors through computation of the q values based on spatially stratified heterogeneity. Meanwhile, a certainty factor (CF) model was used for comparison. The results indicate that the q values of the 12 influencing factors vary obviously, and the dominant factors are also different for the 3 types of landslides, with annual mean precipitation (AMP) being the dominant factor for rockslide distribution, elevation being the dominant factor for rock fall distribution and lithology being the dominant factor for flow-like distribution. Integrating the results of the factor detector and ecological detector, the AMP, annual mean temperature (AMT), elevation, river density, fault distance and lithology have a stronger influence on the spatial distribution of landslides than other factors. Furthermore, the factor interactions can significantly enhance their interpretability of landslides, and the top 3 dominant interactions were revealed. Based on statistics of landslide discrepancies with respect to diverse stratification of each factor, the high-risk zones were identified for 3 types of landslides, and the results were contrasted with the CF model. In conclusion, our method provides an objective framework for landslide prevention and mitigation through quantitative, spatial and statistical analyses in regions with complex terrain.

## Introduction

Landslides have occurred at a high frequency in the southeastern part of the Tibetan Plateau and feature a wide distribution, high activity, large number and large scale; they often cause

(GFZX0404130302), the National Key Research and Development Program of China (2016YFB0501404), and open fund of State key laboratory of resources and environment information system, the innovative research group-geographic spatial-temporal data analysis (08R8B040YA).

**Competing interests:** The authors have declared that no competing interests exist.

road damage, blocking of rivers, casualties and property losses. One of most noticeable examples is the Yigong landslide, which occurred along Zhamunong Creek in 2000 and produced several billion cubic meters of debris, resulting in a landslide dam that intercepted the Yigong River. Dam failure after two months caused heavy casualties and massive property losses downstream of the Yarlu Tsangpo River [1–3]. The Guxiang debris flow, which occurred in 1953, moved $1 \times 10^7$ m$^3$ of debris and resulted in 140 casualties and intercepted the Palong Tsangpo River [4]. Another large-scale glacial/rock fall occurred upstream of the Sedongpu Basin on the left bank of the Yarlu Tsangpo River on October 17, 2018, deposited materials with an estimated volume of $3.1 \times 10^7$ m$^3$ and blocked the Yarlu Tsangpo River for 56 hours [5]. There have also been other severe geological disasters in this region, such as 102 landslides along sections of roads, the Peilong Nongba debris flow, and the Layue landslide, which all incurred heavy losses [6]. Therefore, the National Geological Park of Yigong located in Yigong Township of Bomi County in Tibet is called the "Mountain Disasters Museum of China" [1, 6].

It is important to understand the spatial distribution of landslides by associating environmental factors (or influencing/causative factors) with landslide occurrence [7]. A full study of the spatial association between landslides and environmental factors is of great significance to the evaluation of regional landslide occurrence, slope deformation prediction, spatial distribution and morphological characteristic analysis [8–10]. Reichenbach et al. [11] found 596 unique environmental factors, which can be classified into five clusters, including geological, morphological, hydrological, land cover, and other factors. Altitude, slope angle, slope aspect, distance to rivers, distance to faults, annual precipitation, land use, NDVI, lithology, seismic activity, etc. are causative factors that are commonly used in recent research [7–11]. There has been much research on relationship analyses between environmental factors and landslides, the most frequently used of which include deterministic approaches, statistical analyses, and computational intelligence methods [7, 11]. Deterministic models are typically used in small areas because they require detailed engineering data of soils and rocks, slope geometry and hydrological conditions [12, 13]. Statistical analyses comprise a large proportion, e.g., logistic regression, weight evidence analysis, frequency ratio, weighted linear combination, index of entropy [11, 14, 15]. These methods are simple and widely used, but many problems, such as factor selection, spatial correlation analysis and significance evaluation, have not been solved systematically because of the complex nonlinear characteristics of landslide occurrences. In recent years, there has been a growing trend towards machine learning methods, such as neural network analysis, random forest, naïve Bayes tree, and support vector machine [12, 16–19]. These algorithms are suitable for nonlinear relations among variables and have strong robustness. However, these models are complex, it is difficult to control their internal operations, and they may not produce better results with fewer data features.

As the most typical mountainous area in China, there have been many studies on geological disasters conducted in southeastern Tibet. Through data collection and field investigation, Shang et al. [6] studied the formation mechanism and distribution characteristics of major geological disasters in the northern section of the Great Canyon of the Yarlu Tsangpo River. They believed that the strong tectonic activities, mountain landforms, abundant rainfall, deforestation, road and slope cutting and human activities in this region are mainly responsible for the frequent occurrence of landslides. According to the distribution density of geological disasters and their damage, Liao et al. [20] divided the Nyingchi to Baxoi sections of the Sichuan-Tibet Highway into three subsections and suggested that the geological hazards in the study area show intensive development characteristics, include multiple types and have significant segmentation. Qualitative analysis concludes that loose debris is the material source for landslides, while water-rock actions and river cuttings are the important factors influencing

landslides. From the perspective of climate change, Chen et al. [4] studied the debris flow types, distribution and activity characteristics in the Nyingchi region and concluded that temperature and precipitation changes and different hydrothermal combinations affected the debris flows in the Nyingchi region. Wang [21] investigated the volume and distribution of landslides, rock falls, debris flows and mudslides based mainly on Landsat-7 data through remote sensing survey methods. Topography, geological structures, lithology and precipitation are considered to be the main causes of serious geological disasters in this region.

Although many studies have proceeded on landslide hazards in southeastern Tibet, China, due to the complexity of the relationship between landslides and environmental factors, explicit and systematic explanations are rare. To study the relationship between landslide hazards and reveal the dominant factors and the interactions between these factors, the geographic detector (GeoDetector) method is used in this article. GeoDetector is a new spatial statistical method developed to assess the associations between human disease risks and environmentally feasible risk factors [22–24], and it has already been successfully applied in many fields of natural and social sciences [25–29]. In recent years, it has been proven to be effective in the analysis of landslides and their conditional factors [30–33]. The main goals of this paper are ① to present a landslide inventory of the entire territory of the study area based on multisource images and topographic data and distinguish the major types of landslides and ② to explore the spatial association between different types of landslides and environmental factors and unveil the contributions of various influencing factors based on spatial statistical methods and geographical detectors. Our results are useful for landslide prevention and mitigation in regions of complex terrain such as southeastern Tibet, China.

## Study area and data

### Study area

**Geomorphologic and geographic conditions.** The study area is located in the eastern part of Nyingchi city in the southeastern part of the Tibet Autonomous Region in China and covers parts of Bomi County, Bayi District and Metok County. Its geographic coverage is approximately 94°20′-95°56′E longitude and 29°05′-30°26′N latitude with an area of 15,181.5 km$^2$ [21]. The study area is mainly accessed by highways, with the southern route of the Sichuan-Tibet Highway (G318) running through this region, which is also the intersection of the Sichuan-Tibet Highway, Yunnan-Tibet Highway and construction of the Sichuan-Tibet Railway. This area's location is very important because it serves as a traffic pivot point and is the main access to Tibet. However, as the "*Mountain Disaster Museum of China*", frequent geological disasters have threatened the safety of transportation, residents and property.

The study area is located at the junction of the southeastern and southern Tibetan climate zones and is dominated by subtropical and tropical mountain humid climates, with annual average temperatures of -15–20˚C [34]. Due to large altitude discrepancies and deep canyons, a stereoscopic effect on the climate has formed, which means that the annual mean temperature is higher in the south and in the valley and lower in the north and at the summit. The approximate 4,100 m elevation is the isotherm of the annual mean temperature of 0˚, and most of the mountains experience frost weathering and cryoplanation [35]. Water vapor from the Indian Ocean enters downstream of the Yarlu Tsangpo River and is blocked and condensed in the Nyenchen Tanglha Mountains, resulting in abundant rainfall in the region, with an average annual precipitation reaching 600–1,300 mm [34].

The regional terrain is characterized by high mountains and very steep canyons, with the eastern branch of Nyenchen Tanglha in the north and the northeastern branch of the Himalayas in the south. Namkyabawa is one main peak in the Himalayas and is 7782 m above sea

level, which is the highest point in this area [21] (Fig 1). Many rivers flow through this area, all of which belong to the Yarlu Tsangpo River system [6]. In addition to the Parlung Tsangpo River, the other main tributaries include the Yigong Tsangpo, Lulang and Niyang Rivers [35]. The relative elevation differences between mountains and valleys along these tributaries are generally above 1500 m, with a maximum value above 2800 m, and the slope gradients are generally above 30˚, with a maximum value of 80˚. The water and glacial deposits provide abundant material sources for landslides and debris flows [21]. The hillsides are covered with forest and grassland, and populated residential areas and farmland are scattered in narrow valleys.

**Geological environment.** The study area is located in the subduction zone of the Indian and Eurasian Plates [21, 36], and the Yarlu Zangbo River is developed along the suture zone. The early Proterozoic crystalline basement mainly occurs in Nyenchen Tanglha, and the main lithologies are schist, gneiss, marble, metamorphic rock, etc. Limestone, marble, slate, conglomerate and sandstone of the Devonian, Carboniferous and Jurassic and a large area of Yanshanian granite are developed in the Parlung Tsangpo Basin, and modern glaciers exist in the northeastern part of the study area. The trending direction of tectonic lines are NWW- in most places, and there are SW-, NW-, SE- and SN-oriented faults around the Yarlu Canyon. The Jiali-Ranwu and Bomi Faults, which run through the east and west of this area, have a range of influence of up to 30 km [37] and are closely related to neighboring geological disasters.

Due to its straddling a plate boundary, the study area is an earthquake-prone region. According to the records of the *Earthquake Disasters Program of the United States Geological Survey* (https://earthquake.usgs.gov/), there were 134 earthquakes with magnitudes above 4.5 in this region from 1950 to 2019 (Table 1). Most of the earthquakes are concentrated in a circular area in the northwestern part of the region and north of Yigong Lake.

## Data

**Landslide inventory.** Translational and rotational rockslides, rock falls, and debris flows are the main types of geological disasters, which in total account for 92% of the geological disasters identified in the southeastern part of the Tibetan Plateau [1]. Wang [21] pointed out that translational and rotational rockslides, rock falls, debris flows and mudslides are geological disasters with wide distributions, strong activities, large quantities and large scales in the study area. They adopted a remote sensing survey method based on Landsat-7 data and mapped a total of 406 mass movements in this area, including 224 large rockslides, 50 rock falls, 41 large debris flows, 86 V-shaped mudslides and 5 slope mudslides. The remote sensing survey method is efficient for identifying catastrophic landslides after heavy storms or high-intensity earthquakes, which means that shortly after sliding, there are obvious marks in remote sensing images [38–40]. However, shadows and low resolution in single optical images hinder landslide detection, especially for old landslide recognition. Therefore, a combined multisource image and topographical data method is adopted for landslide identification in our study. GF-1 images with 16-meter resolution, GF-2 images with 2-meter resolution and Landsat-8 fusion images with 15-meter resolution are used for landslide interpretation, and SRTM-based digital elevation models (DEMs) are used for visual interpretations of landslides based on their morphometric context.

Based on remote sensing and geomorphologically recognizable characteristics, we simplified combined translational and rotational rockslides into a series of rockslides, debris flows, mudslides and complex landslides, which were simplified into a series of flow-like landslides and rock falls. In this paper, the term landslide is used to indicate all 3 types of mass movements mentioned above. The new landslides are easily identified in optical images, and the old

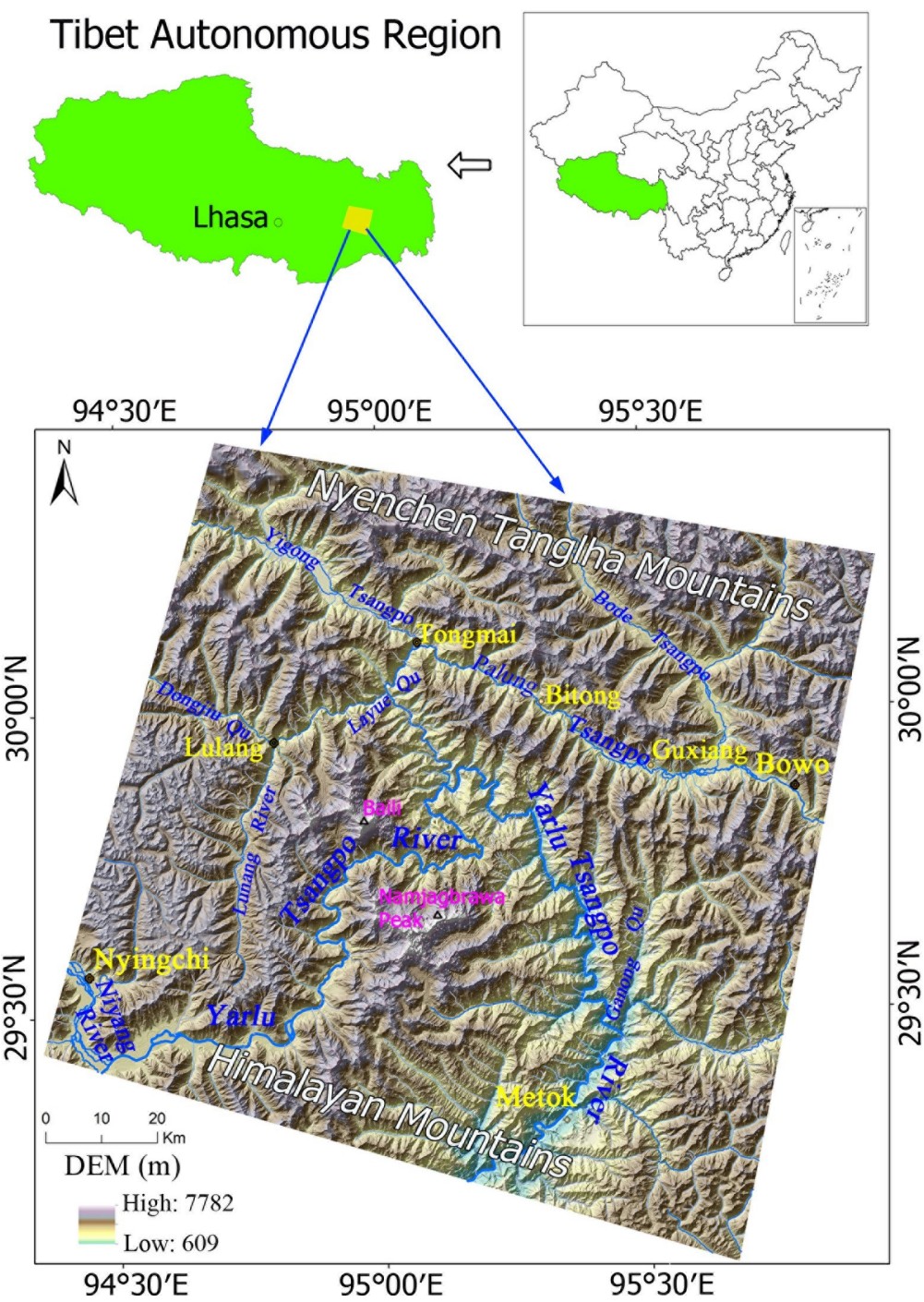

**Fig 1. Map of the study area.**

**Table 1. Statistics of earthquakes with $M_L \geq 4.5$ in the study area.**

| Magnitude, $M_L$ | 4.5~5 | 5~6 | $\geq 6$ |
|---|---|---|---|
| Times | 117 | 16 | 1 |

landslides can be found by combining hillshade, slope and contour maps derived from the DEM with optical images. Therefore, new rockslides (Fig 2B), flow-like landslides (Fig 2D) and rock falls (Fig 2F) are identified in optical images, and old rockslides (Fig 2A) and flow-like landslides (Fig 2C) are identified by combining slope, relief and hillshade maps derived from the DEM with optical images. The image features we used for the 3 types of mass movement detections are main scarps, counterscarps, deflected drainages, accumulation areas, source areas, flow areas and flow lobes (Fig 2). Google Earth and some studies [5, 6, 21, 51] were adopted to assist in the determinations, and the interpretations along the G318 highway were also edited through field investigations. The Qinghai-Tibet Plateau has unique topographical landscape and scientific research significance, and the Chinese Academy of Sciences and other scientific research institution in China carries out various scientific research investigations here every year. Our Field work obtained the permission from our work units and local government.

A complete landslide inventory is essential to understand the mechanism and characteristics of landslides in a region [41]. Landslides in the study area were primitively mapped by Lv et al. [42] in 2002 and Wang [21]. Although once comprehensive source material about the spatial distribution and type of landslides in the study area, we found this inventory to be inaccurate and incomplete in some places (Fig 3). This may be because the main purpose of that study focused on the impact of geological disasters on traffic into Tibet and the use of single and low-resolution data resources. Therefore, for the purpose of studying the relationship between landslides and environmental factors, a new landslide inventory is formed based on our methods. A total of 1766 large landslides were identified in the study area, including 698 rockslides, 343 rock falls, and 725 flow-like landslides.

**Environmental factors.**   A total of 12 environmental variables (factors) are used for spatial association analysis in our study, and their basic data are all available in the public domain and may be correlated with geological disasters according to previous studies [11, 43–45]. The Shuttle Radar Topography Mission (SRTM) and earthquake data are collected from the *Land Processes Distributed Active Archive Center* (https://lpdaac.usgs.gov/) and the *Earthquake Disasters Program of the USGS*. The geological data are collected from the *National Geological Archives of the China Geological Survey* (http://ngac.org.cn/). Meteorological data are acquired from the *Earth Big Data Science and Engineering Data Sharing Service System of the Chinese Academy of Science* (http://data.casearth.cn/) [46], and land use and land cover data are collected from *the National Geomatics Center of China* (http://www.globallandcover.com/GLC30Download/index.aspx). All these data have passed stringent quality validation and are reliable, which has been verified by many public papers and reports [47, 48].

The 12 environmental factors considered in this paper can be divided into geological factors (lithology and fault distance), earthquake factor (epicenter distance), topographic factors (elevation, slope, and aspect), hydrological factors (river distance and river density), meteorological factors (annual mean precipitation (AMP) and annual mean temperature (AMT)), and land use and land cover (vegetation fractional cover (VFC) and land cover). Details of these factors are shown in Table 2.

The independent variables must be categorical (discrete) type data when performing a spatial analysis based on geographical detectors, in which continuous variables need to be converted into discrete variables. The elevation data were produced from SRTM-1 with a spatial resolution of 30 m, which was divided into 10 levels according to the equidistance (Fig 4A). According to natural breaks, the slope data were generated from SRTM-1 and divided into 9 levels (Fig 4B). The aspect data, which were divided into 10 types, were also generated from SRTM-1 at the same spatial resolution (Fig 4C). The aspect factor plays an important role in mountain ecology through its influence on sunshine duration and solar radiation intensity,

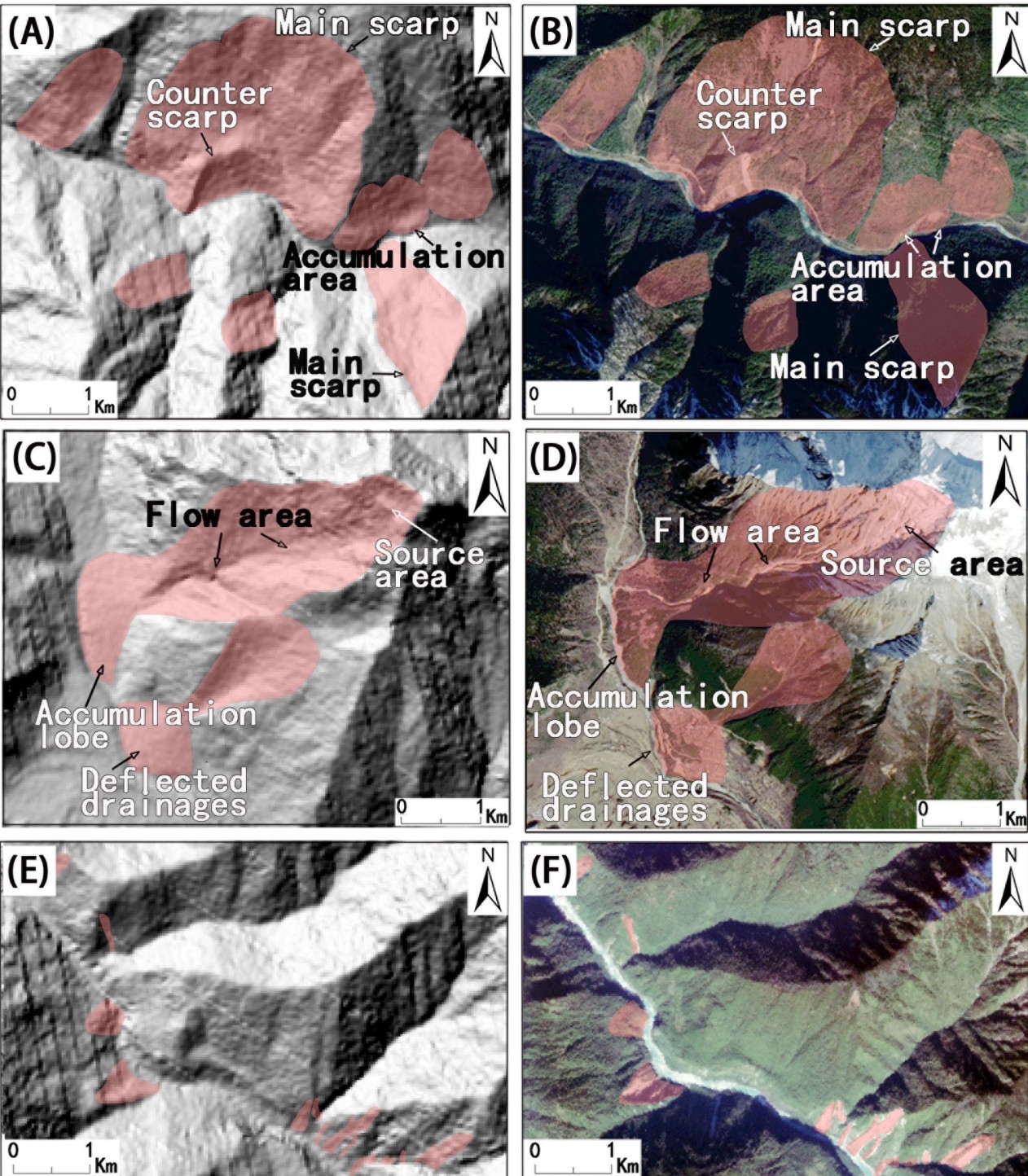

**Fig 2. Examples of different types of landslides in our study area and their features.** (A) Rockslides and their diagnostic features on a DEM image, (B) rockslides and their diagnostic features on a Landsat-8 image, (C) flow-like landslides and their diagnostic features on a DEM image, (D) flow-like landslides and their diagnostic features on a Landsat-8 image, (E) rock falls along the river on a DEM image, and (F) rock falls along the river on a GF-1 image.

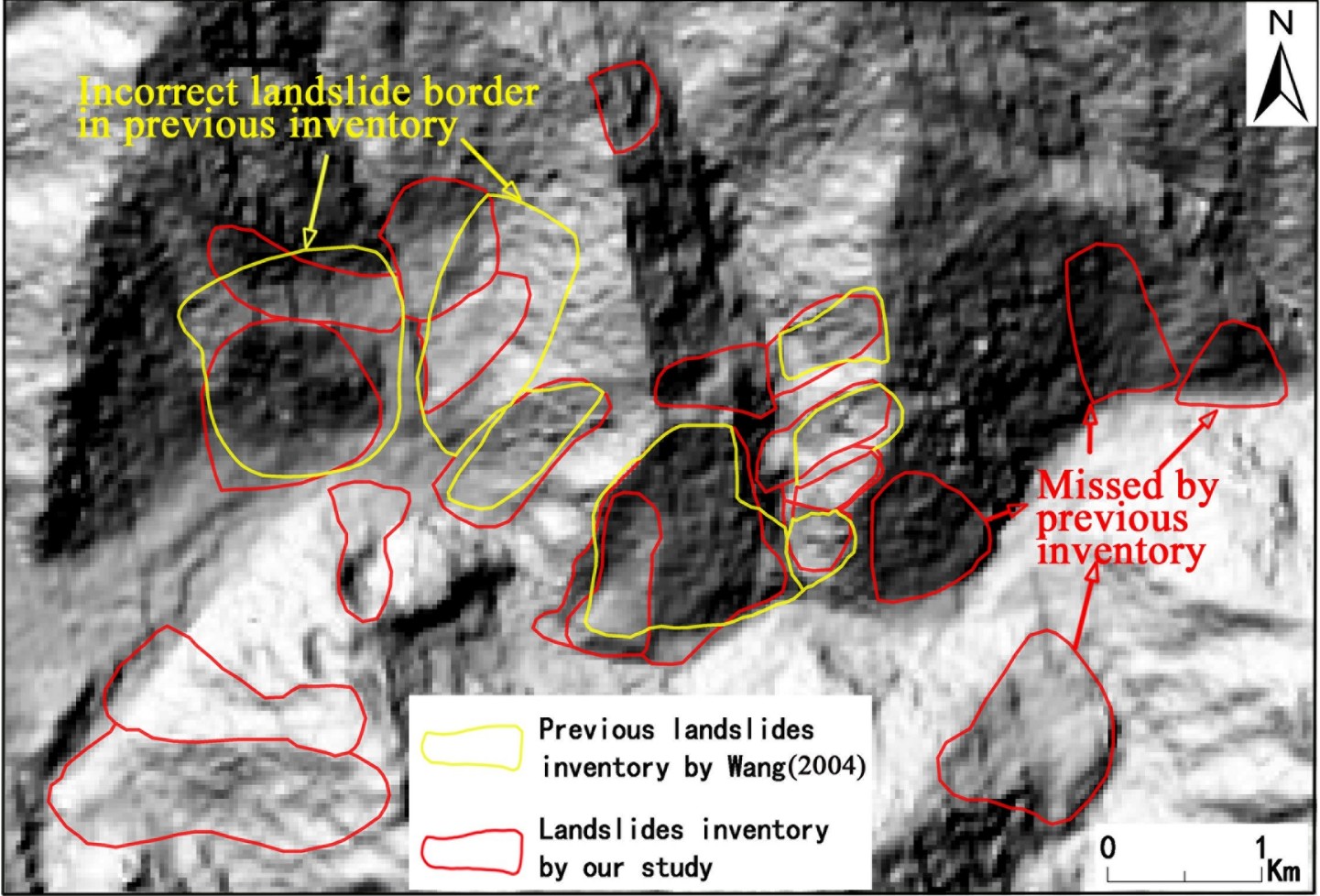

**Fig 3. Comparison of landslide inventories obtained by our methods and previous landslide inventories by Wang [21].**

**Table 2. Information on landslide points and influencing factors.**

| Variables | Name | Original Data Structure | Variable Type | Data Description | Class |
|---|---|---|---|---|---|
| Y | Landslide points | point | continuous | The center point of landslide, debris flow or rock fall source area | Landslides |
| X1 | Fault distance | Line | continuous | Distance to the fault | Geological |
| X2 | Lithology | polygon | discrete | Different types of rock masses | Geological |
| X3 | Epicenter distance | point | continuous | Distance to the epicenter | earthquake |
| X4 | Elevation | Raster | continuous | SRTM-1 DEM with 30 m resolution | Topographic |
| X5 | Slope | Raster | continuous | Derived from DEM | Topographic |
| X6 | Aspect | Raster | continuous | Derived from DEM | Topographic |
| X7 | River distance | Line | continuous | Derived from river line | Hydrological |
| X8 | River density | Line | continuous | Derived from river line | Hydrological |
| X9 | AMP | Raster | continuous | Annual mean precipitation | Meteorological |
| X10 | AMT | Raster | continuous | Annual mean temperature | Meteorological |
| X11 | VFC | Raster | continuous | Derived from the remote sensing images | Land cover and land use |
| X12 | Land cover | Raster | discrete | Derived from the remote sensing images | Land cover and land use |

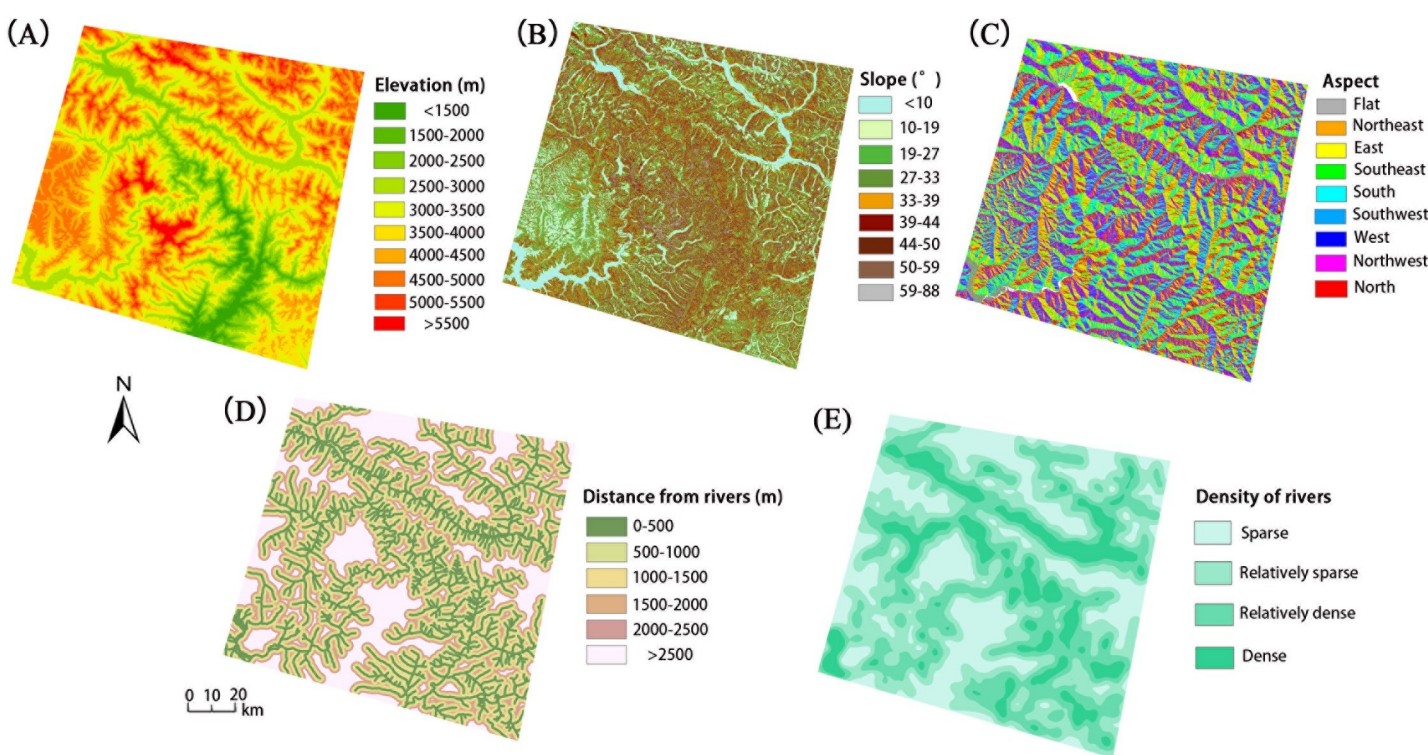

**Fig 4.** Topographic and hydrological factors, including (A) elevation, (B) slope, (C) aspect, (D) distance from river, and (E) density of river.

which may influence rock weathering. The digitized river system in the study area was mainly acquired from SRTM-1 and partially digitized from remote sensing images as a supplement. The distance to rivers was divided into 6 levels, as shown in Fig 4D. The river density was also acquired based on a digitized river system using the kernel density tool provided by ARCGIS10. Four types of river densities were determined in the study area: sparse, relatively sparse, relatively dense and dense (Fig 4E). River erosion is an important factor in mass movements, and the loose deposits in the watershed also provide sufficient materials for the occurrence of landslides.

Lithology is believed to be related to landslides in similar external environments. The engineering properties (e.g., hardness and weathering ability) of the rock mass in which the landslides occur vary greatly [49, 50]. The lithology data in our study area were digitized from 1:500,000 geological maps and related references [51, 52] and included 9 categories (Fig 5A). The main type of lithology in the study area is Presinian deep metamorphic rock, and Miocene granite is also widely distributed in the eastern and northeastern parts of the study area. Faults are an important factor that influence the spatial distribution of landslides. Shang et al. [6] pointed out that the major fault zone often controls the location and scale of the landslide. The faults in our study area were obtained from 1:500,000 geological maps combined with the remote sensing interpretation of linear features. There are a large number of landslides and rock falls along some faults, and the displacement field caused by fault activities has destructive effects on the slopes [37, 53]. The distances to faults were divided into 6 levels, as shown in Fig 5B, as the influence of the fault can simply be determined by the distance in a similar geological environment. Earthquakes usually induce mass movement directly and cause slope instability [49, 53–56]. The epicentral distance of the earthquake is positively correlated with the landslide. The Euclidean distance to the epicenter in our study area is divided into 7 levels by natural breaks (Fig 5C).

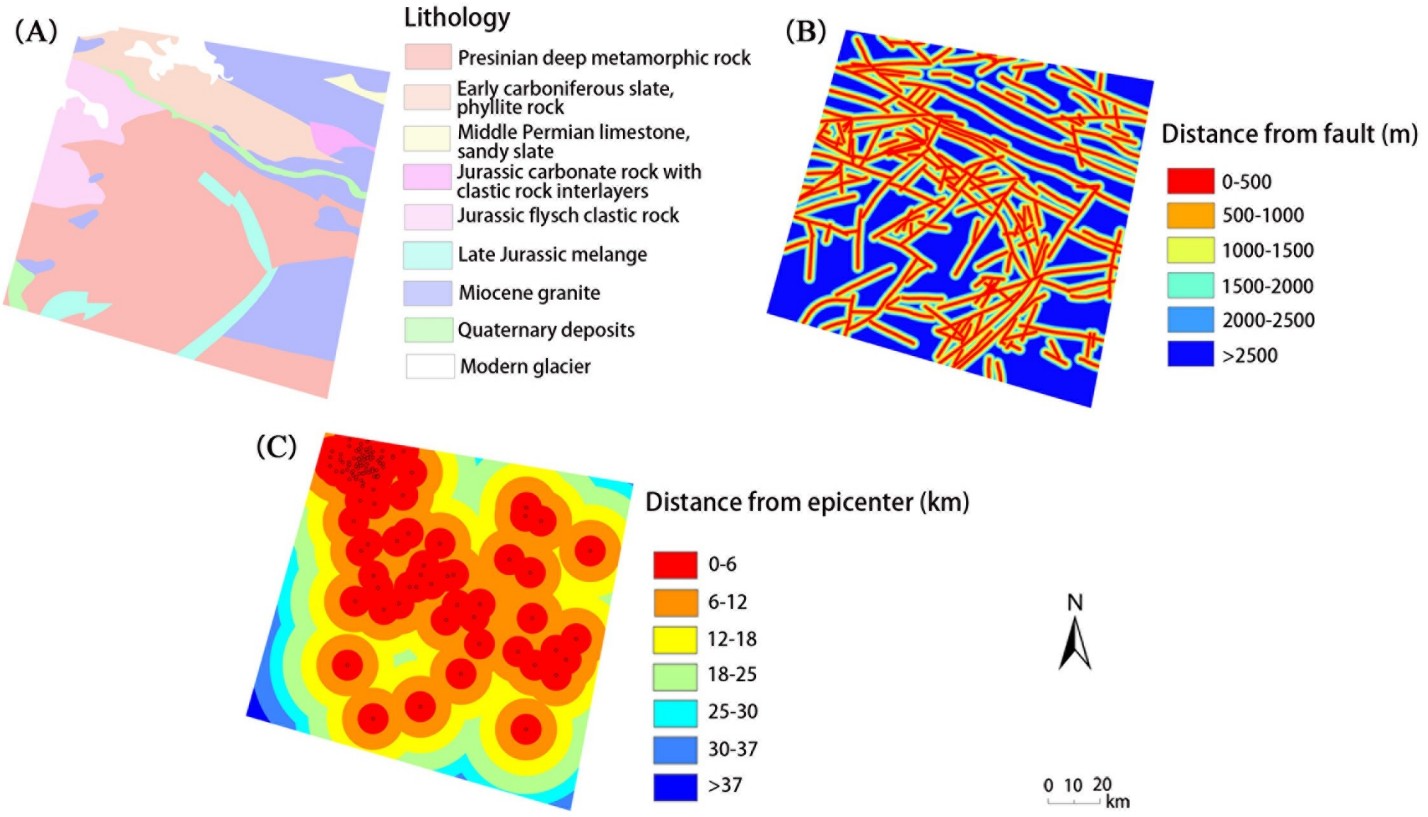

**Fig 5. Geological influencing factors, including (A) lithology, (B) distance from fault, and (C) distance from epicenter.**

The mean values of annual mean temperature (AMT) and annual mean precipitation (AMP) grid data with a 1 km resolution ranging from 1980 to 2000 were calculated [46] and then divided into 6 levels (Fig 6A) and 5 levels (Fig 6B), respectively, by manual breaking using

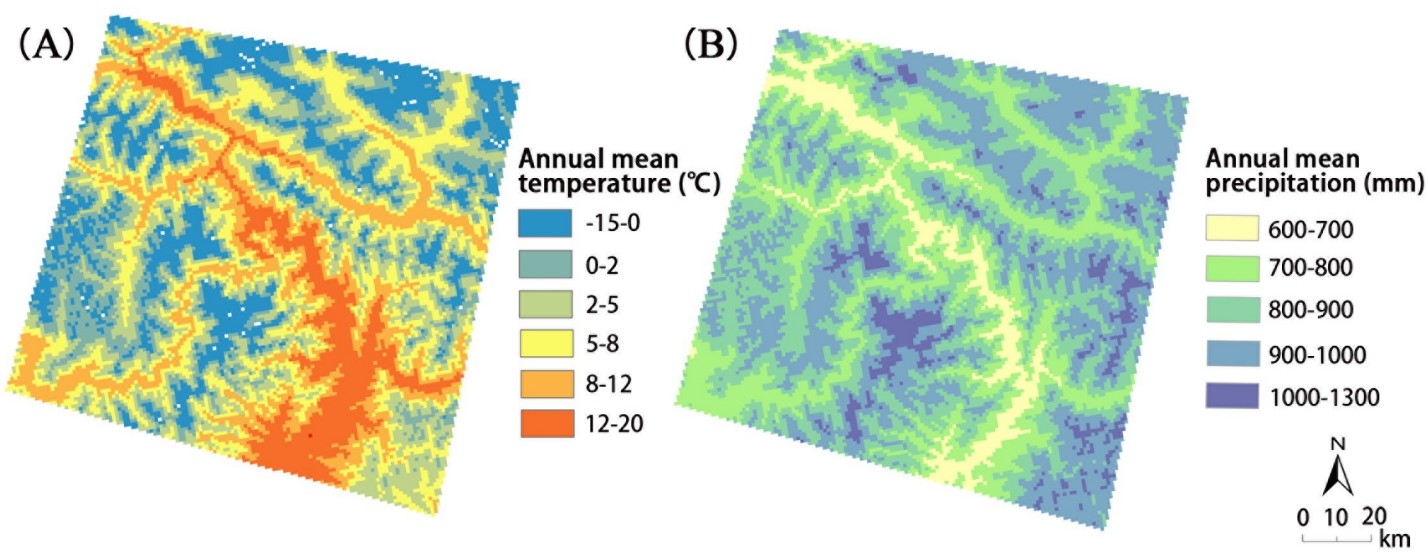

**Fig 6.** Meteorological factors, including (A) Annual Mean Temperature (AMT) and (B) Annual Mean Precipitation (AMP).

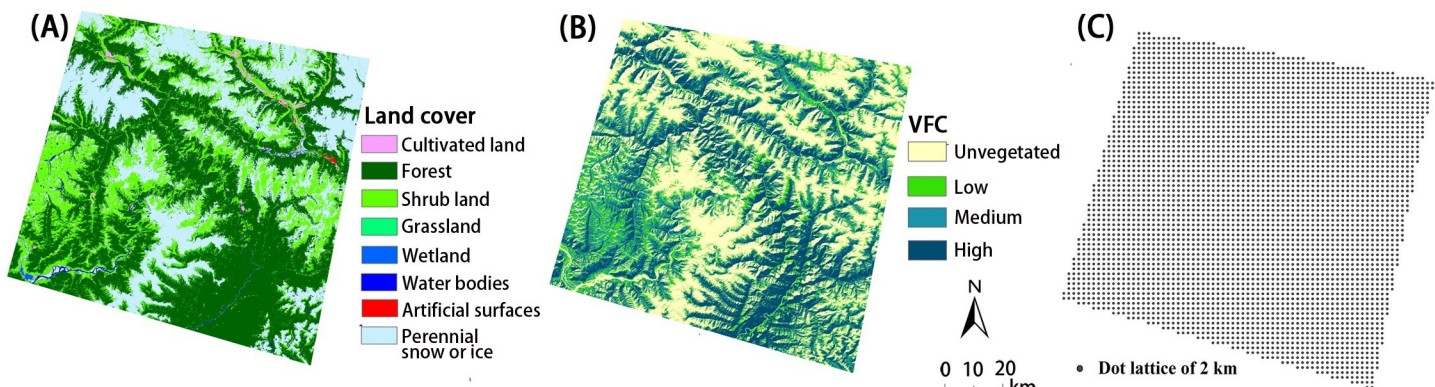

**Fig 7. The influencing factors of land use and land cover for geological disasters.** (A) Land cover, (B) vegetation fractional cover (VFC), and (C) sampling points of the entire region.

the meteorological influencing factor. Snowmelt, change in the cover and rock weathering caused by temperature change are closely related to landslides [3–6], and rainfall is an important triggering factor for debris flows and other types of landslides [4, 6].

The land use and land cover data were certified by the National Geomatics Center of China as the global geo-information public product-Globeland30 [47] and were classified as cultivated land, forest, grassland, shrub, wetland, water bodies, artificial surfaces, and permanent snow and ice, totaling 8 categories in our study area (Fig 7A). The vegetation fractional cover (VFC) data were extracted from the Landsat 8 data in ENVI and then divided into 4 levels according to the following division criteria: VFC<0.2 as unvegetated, 0.2–0.5 as low vegetated, 0.5–0.8 as moderately vegetated and 0.8–1 as highly vegetated (Fig 7B).

All raster data were resampled to a 30 m resolution to ensure the consistency of the spatial resolution among different environmental factors. The fishnet function of ARCGIS10 was applied to obtain evenly distributed points in the entire region with a 2 km interval, achieving a total of 3,795 points (Fig 7C). The values of the 12 influencing factors and the landslide density data were then extracted as the operational data for the geographic detector software.

## Method

### GeoDetector

The geographic detector method as a spatial statistical method can be used for detecting spatial variations and revealing driving factors [22–24]. In GeoDetector, the independent variable $X$ is taken as the environmental (influencing) factor that contributes to the spatial distribution of geological disasters, and the dependent variable $Y$ is introduced to represent geological disasters. We use kernel density to calculate $Y$, as the original data are center points in the source regions of landslides. Because GeoDetector works with the independent variable $X$ as a categorical variable, the variable $X$ must either be a categorical layer or needs to be partitioned. The lithology, watershed and land use data are already categorical variables in our dataset, and other continuous variables are partitioned by natural, manual or equidistance breaks (Table 2). The GeoDetector software developed in Excel is freely available at http://www.GeoDetector.cn/ [24].

The key assumption of the GeoDetector method is listed as follows: if the independent variable $X$ has an important influence on the independent variable $Y$, then they would represent a similar spatial distribution [24, 27]. Spatial variations and factor analysis are measured by the $q$

values acquired from GeoDetector. There are four modules in GeoDetector: the factor detector, the risk detector, the ecological detector and the interaction detector [24].

*A factor detector* is used to detect the spatial heterogeneity of the dependent variable $Y$ and the explanatory power of the independent variable $X$ to the dependent variable $Y$, which is measured by the power of the derived $q$ value, $q \in [0,1]$. The equations are listed below:

$$q_X = 1 - \frac{\sum_{h=1}^{L} N_h \sigma_h^2}{N\sigma^2} = 1 - \frac{SSW}{SST} \tag{1}$$

$$SSW = \sum_{h=1}^{L} N_h \sigma_h^2, SST = N\sigma^2 \tag{2}$$

where $h = 1, 2,.., L$ is the number of categories of the independent variable $X$, $N_h$ is the unit number in zone h, $N$ is the unit number in the whole region, $\sigma_h^2$ is the variance of $Y$ in zone h, and $\sigma^2$ is the global variance of $Y$ in the whole region. SSW represents the weighted sum of the local variance (weighted by the number of samples in each zone), and SST is the global variance. The definition equations of $\sigma_h^2$ and $\sigma^2$ are as follows:

$$\sigma_h^2 = \frac{1}{N_h} \sum_{i=1}^{N_h} (D_{h,i} - \overline{D_h})^2 \tag{3}$$

where $D_{h,i}$ is the value of the $i_{\text{th}}$ unit of $Y$, and $\overline{D_h}$ is the average of D in subsegmented zone $h$.

$$\sigma^2 = \frac{1}{N} \sum_{j=1}^{N} (D_j - \overline{D})^2 \tag{4}$$

In there, $D_j$ is the value of the $j_{\text{th}}$ unit of the whole study area and $\overline{D}$ is the global average of D in the entire study area.

From Eq (1), we know that $q_X$ is inversely related to the ratio of the weighted sum of the local variance (SSW) to the global variance (SST). In an ideal situation, if the environmental factor $X$ entirely controls the spatial distribution of $Y$, the $SSW$ is 0 and $q_X = 1$. In contrast, if the environmental factor $X$ is completely unrelated to the spatial distribution of $Y$, the $SSW$ is the same as the $SST$, and $q_X = 0$. Above all, the factor detector with $q_X$ as the proportion of the spatial variation of $Y$ is explained by the environmental factor $X$. The larger the $q_X$, the stronger the interpretability of factor $X$ on the spatial distribution of $Y$. Note that the influencing factor assesses the spatial association degree between $X$ and $Y$, which does not explain their causal relationship.

*The risk detector* can determine whether there exist significant discrepancies in the density of landslides between zones with different influencing factors $X$ and identify the high-risk zones of landslides.

*The ecological detector* compares the differences in the effects of various environmental factors on landslides, and the F test is adopted for this determination. Through computation and comparison of the $q$ values of two different environmental factors $X_1$ and $X_2$, $q_{X1}$ and $q_{X2}$, with the $q$ value of a new factor is created by overlaying factors $X1$ and $X2$ as $q_{X1 \cap X2}$.

*An interaction detector* can determine whether there exists an interaction between the two factors, the intensity of the interaction, and the directionality of this relationship: linear or nonlinear. It may detect the multiplication and any other relationship of the two-factor superposition. If $q_{X1 \cap X2} > q_{X1}$ and $q_{X1 \cap X2} > q_{X2}$, it indicates that the varieties may enhance each other; if $q_{X1 \cap X2} > q_{X1} + q_{X2}$, the varieties may enhance each other in a nonlinear relationship; if

**Table 3. Types of interaction between two covariates.**

| Criterion | Type of interaction |
|---|---|
| $q_{X1 \cap X2} < Min(q_{X1}, q_{X2})$ | Nonlinear weakened |
| $Min(q_{X1}, q_{X2}) < q_{X1 \cap X2} < Max(q_{X1}, q_{X2})$ | Univariate nonlinear weakened |
| $q_{X1 \cap X2} > Max(q_{X1}, q_{X2})$ | Bivariate enhancement |
| $q_{X1 \cap X2} = q_{X1} + q_{X2}$ | Independent |
| $q_{X1 \cap X2} > q_{X1} + q_{X2}$ | Nonlinear enhancement |

Note: $X1$ and $X2$ represent two different environmental factors of landslides.

$q_{X1 \cap X2} < Min(q_{X1}, q_{X2})$, the varieties may weaken each other in a nonlinear relationship; if $q_{X1 \cap X2} = q_{X1} + q_{X2}$, then they are independent from each other (see Table 3).

## Certainty factor model

The certainty factor (CF) model is based on a probability function, which was first proposed by Shortliffe and Buchanan [57] and later modified by Heckerman [58]. The CF model has been applied by different researchers in susceptibility analyses of landslide triggering factors [10, 59–61]. The function of the CF model is as follows:

$$CF_{ij} = \begin{cases} \dfrac{f_{ij} - f}{f_{ij}(1 - f)}, & \text{if } f_{ij} \geq f \\[2ex] \dfrac{f_{ij} - f}{f(1 - f_{ij})}, & \text{if } f \geq f_{ij} \end{cases} \tag{5}$$

where $f_{ij}$ is the conditional probability that having a number of landslides happening in layer $i$ of factors $j$ and $f$ is the prior probability of having all the landslides occurring in the whole area; the result, $CF_{ij}$, is the certainty factor given to a certain layer $i$ of factor $j$.

The value of the certainty factor $CF_{ij}$ ranges from -1 to 1. A positive value represents the increase in the certainty of an event occurrence, which means that the certainty of landslide occurrence is high, while the negative value represents the decrease of an event certainty, indicating that the certainty of landslide occurrence is low. The value of a certainty factor close to 0 represents that the prior probability is close to the conditional probability, and it is impossible to determine the certainty of landslide occurrence, which means this layer cannot determine whether it is a landslide prone area [10].

According to the statistical relationship between landslide events and environmental factors [59–61], the CF model could be used to determine the high-risk zones and the key factors of landslides occurrence. The results of the CF model are compared with GeoDetector to ensure the spatial association analysis between landslides and environmental factors.

## Results and discussion

### Landslide inventory and their spatial distribution

A total of 1766 total landslides were identified in the study area, including 698 rockslides, 343 rock falls, and 725 flow-like landslides. The locations of landslides are the locations of the center points of the source regions (Fig 8). The landslides that occurred in the study area are characterized by intense, periodic, repeatable and grouped landslides [5, 21], which indicate that the distribution of the landslides is spatially heterogeneous. Spatial heterogeneity means the uneven distribution of a trait, event, or relationship across a region [62]. According to the

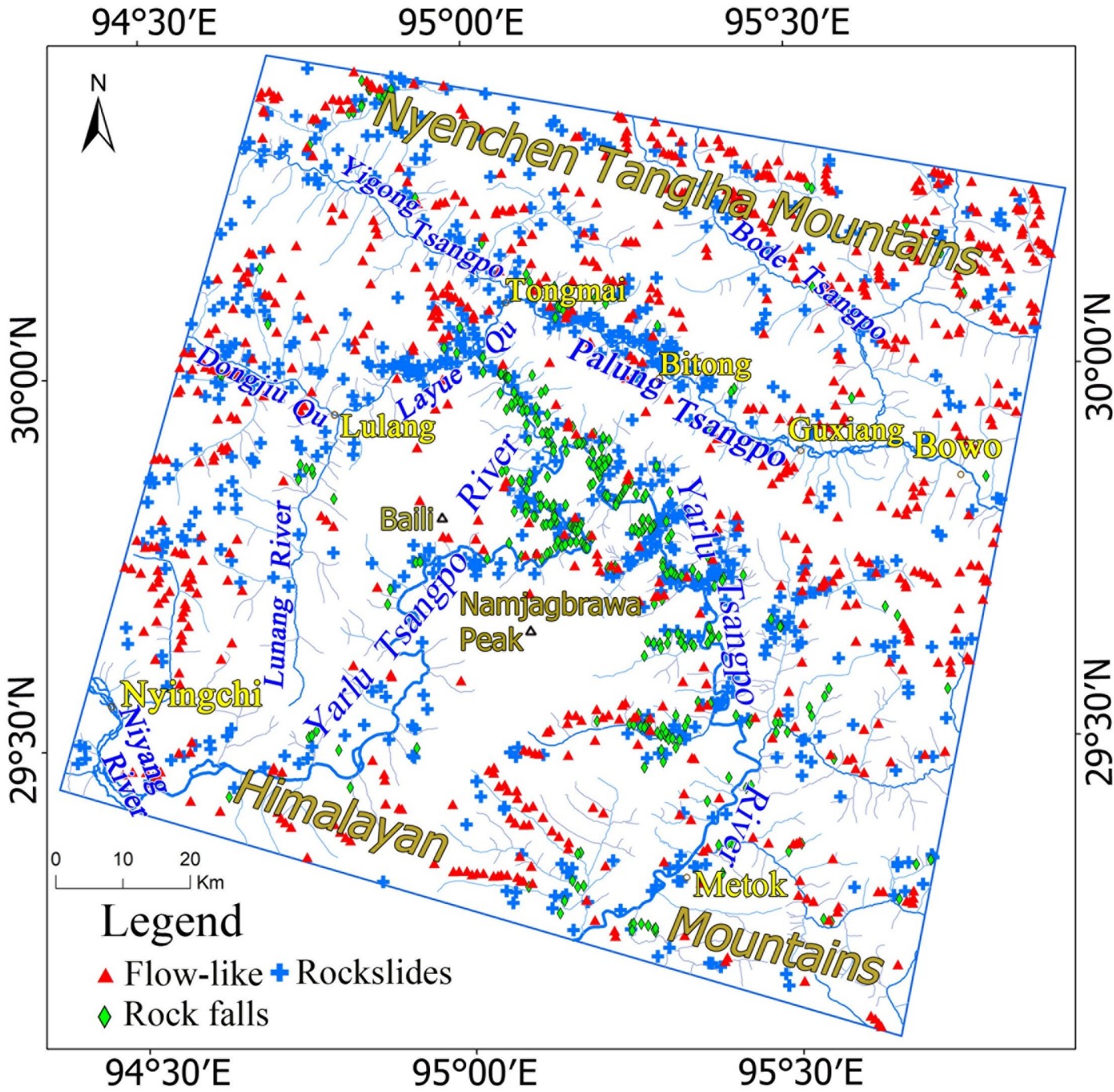

**Fig 8. Landslides inventory map in the study in the southeastern part of the Tibetan Plateau.** Red triangles represent flow-like landslides, green diamonds represent rock falls, and blue crosses represent rockslides.

inventory, the northwestern part of the study area and the great canyon of the Yarlu Tsangpo River are the areas with the highest concentrations of landslides, such as the Layue Qu River-banks, the Bitong to Tongmai section of Parlung Tsangpo, upstream of Dongjiu Qu, and the neighboring and eastern areas of the Great Canyon of the Yarlu Tsangpo River, which

**Table 4. The *q* values of the influencing factors with respect to different types of landslides.**

| Factor No. | Rockslides | Rock falls | Flow-like landslides | Total landslides |
|---|---|---|---|---|
| X1 | 0.0603 | 0.0211 | 0.0098 | 0.0641 |
| X2 | 0.0393 | 0.0603 | 0.0729 | 0.0245 |
| X3 | 0.0180 | 0.0400 | 0.0142 | 0.0306 |
| X4 | 0.1119 | 0.1711 | 0.0228 | 0.1132 |
| X5 | 0.0098 | 0.0175 | 0.0070 | 0.0219 |
| X6 | 0.0101 | 0.0035 | 0.0130 | 0.0136 |
| X7 | 0.1010 | 0.0611 | 0.0295 | 0.0860 |
| X8 | 0.0522 | 0.0326 | 0.0117 | 0.0548 |
| X9 | 0.1604 | 0.1489 | 0.0194 | 0.1549 |
| X10 | 0.1164 | 0.1668 | 0.0174 | 0.1167 |
| X11 | 0.0341 | 0.0181 | 0.0046 | 0.0300 |
| X12 | 0.0356 | 0.0380 | 0.0086 | 0.0291 |

Note: X1 represents the fault distance, X2 represents the lithology, X3 represents the epicenter distance, X4 represents the elevation, X5 represents the slope, X6 represents the aspect, X7 represents the river distance, X8 represents the river density, X9 represents the AMP, X10 represents the AMT, X11 represents the VFC, and X12 represents the land cover.

contribute 58% of the landslides. The others are usually scattered along the rivers in other parts of the study area.

The concentration areas of the 3 different types of landslides varied. The rockslides are concentrated in the Layue Qu riverbanks, the neighborhood of Bitong County and Tongmai County, and the eastern area of the Great Canyon. The new recognized rock falls are concentrated in the downstream area of Parlung Tsangpo and in the neighborhood of the Great Canyon of Yarlu Tsangpo River, which may be caused by an Ms 6.8 earthquake that occurred nearby in November 2017. Flow-like landslides are distributed along rivers and their tributaries, which are scattered throughout the study area. Due to the significant differences in topography, geological environments and climatic distribution in the study area, the spatial differentiation of different types of geological disasters is obvious.

## Quantitative analysis of the spatial association between geological disasters and environmental factors

**The dominant factors and interactions of the environmental factors.** To detect the individual and combined effects of environmental factors on different types of landslides, a geographical detector analysis was used. The results show that the dominant factors and the interaction of the environmental factors on different types of landslides are not the same, and the explanatory power (*q* value) of the combined factors is higher than that of a single factor. We use the word "dominant" to indicate that the environmental factor or interaction has the largest explanatory power (q value).

*(1) Landslides.* According to a previous study, intense water-rock interactions and temperature, rainfall, river erosion, physical weathering, etc. are important factors for the occurrence of landslides in the study area [3, 20]. In the running results of the factor detector, AMP, AMT, elevation, river density and fault distance contribute more to landslide occurrence than other environmental factors. The influencing sequence of these factors by the *q* value is AMP >AMT > elevation > river density> fault distance, which means that the dominant factor for the landslides is AMP (*q* = 0.1549) (Table 4, Fig 9). The ecological detector of GeoDetector shows that the influences of the environmental factors, including AMP, AMT, elevation, river

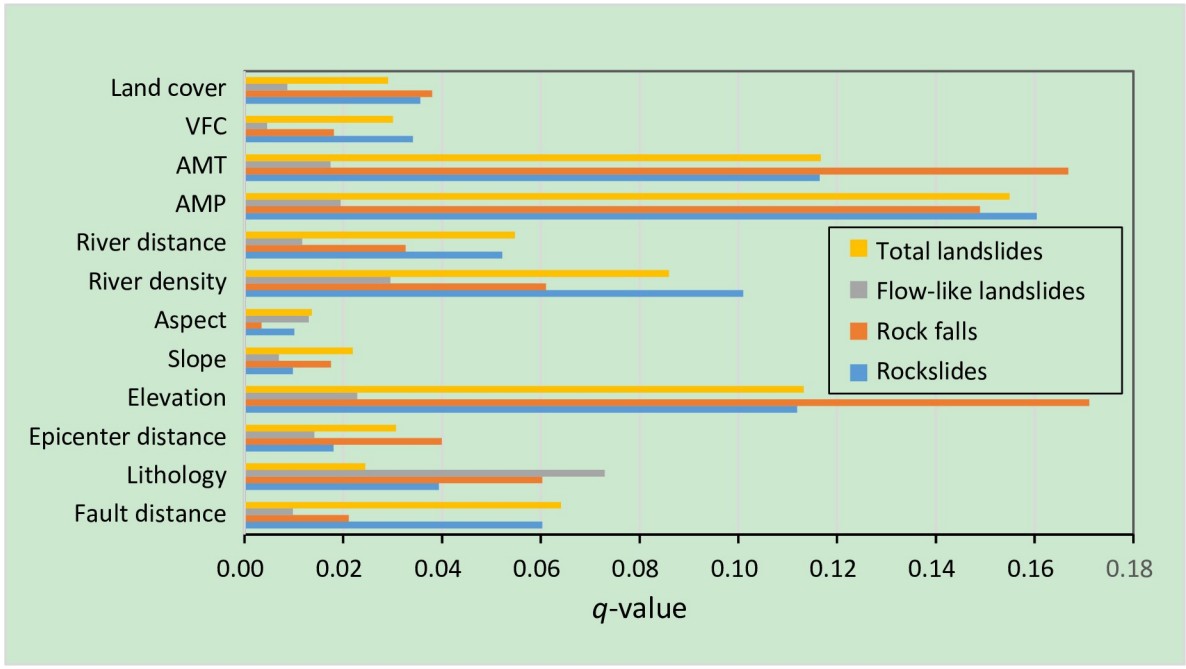

**Fig 9. The *q*-statistic indices of different types of landslides calculated by GeoDetector.** A graphic representation of the relative contributions of environmental factors to their formation.

density and fault distance on the spatial distribution of landslides are significantly different from those of other factors.

The interaction detector results show that the interaction of environmental variables enhances the influence on landslides. The interaction modes with the top 3 explanatory powers were recorded, and the statistical results are shown in Table 5. The dominant interaction factors for landslide distribution are the lithology and AMP (q = 0.2365), which means that in areas with different lithologies and annual mean precipitation values, the incidence of landslides varies greatly. The interaction between lithology and epicenter distance (q = 0.2217) and the interaction between lithology and AMT (q = 0.2211) also play a great role in landslide distribution. Although lithology has a low q value, it is also important because of the interaction of lithology and AMP and epicenter distance, and AMT can explain over 22% of the spatial distribution of landslides.

*(2) Rockslides*. For the rockslide type, AMP was also the dominant factor (*q* = 0.1604) (Table 4, Fig 9). The environmental factors of AMT, elevation, river density and fault distance have higher explanatory power than others. Ecological detectors show that AMP, AMT, elevation, river density and fault distance are significantly different from the others.

**Table 5. The dominant interactions between two environmental factors in different types of landslides.**

|  | Rockslides | Rock falls | Flow-like landslides | Total landslides |
|---|---|---|---|---|
| **Top 1 Dominant interaction (q) and type** | X2∩X9 (0.2201), Nonlinear enhancement | X3∩X4 (0.2619), Nonlinear enhancement | X2∩X4 (0.1292), Nonlinear enhancement | X2∩X9 (0.2365), Nonlinear enhancement |
| **Top 2 Dominant interaction (q) and type** | X1∩X9 (0.2069), Nonlinear enhancement | X2∩X4 (0.2595), Nonlinear enhance | X2∩X10 (0.1269), Nonlinear enhancement | X2∩X3 (0.2217), Nonlinear enhancement |
| **Top 3 Dominant interaction (q) and type** | X2∩X7 (0.2021), Nonlinear enhancement | X2∩X9 (0.2568), Nonlinear enhancement | X2∩X9 (0.1047), Nonlinear enhancement | X2∩X10 (0.2211), Nonlinear enhancement |

Note: Variables X4, X5, X6, X8, X9, X10, etc. in this table have the same meanings as those in Table 4.

Similar to the results of total landslides, the explanatory powers of combined factors on rockslides are stronger than those of individual factors. The results show that the top 3 dominant interactions for rockslides are the interaction between lithology and AMP (0.2201), the interaction between fault distance and AMP (0.2069), and the interaction between lithology and river density (0.2021).

In conclusion, AMP, AMT, river density and fault distance are important factors on rockslides, which is a similar to that of previous studies in this area [1–6, 20, 21]. Although lithology has a low *q* value, it is also important because of the interaction of lithology and AMP, and the interaction of lithology and river density can explain over 20% of the distribution of rockslides.

*(3) Rock falls*. The situation is different for the rock fall type, for which elevation is the dominant factor (q = 0.1711); the environment factors of AMT, AMP, river density and lithology have a stronger influence on the spatial distribution of rock falls with the sequence of AMT > AMP > river density > lithology. Ecological detectors show that AMP, AMT, elevation, river density and lithology are significantly different from the other factors.

The top 3 dominant interactions for rock falls are the interaction between epicenter distance and elevation (0.2619), the interaction between lithology and elevation (0.2595), and the interaction between lithology and AMP (0.2568). Although epicenter distance has a low *q* value, it is important because the interaction of epicenter distance and elevation can explain over 26% of the distribution of rock falls. This result is consistent with the phenomenon found in our inventory, which is that many of the rock falls are caused by an Ms 6.8 earthquake that occurred nearby in November 2017. In addition, lithology is an important factor, which has been proven in many previous studies [49, 50].

*(4) Flow-like landslides*. Contrary to conventional viewpoints, lithology is the dominant factor for the distribution of flow-like landslides (q = 0.0729), and no other environmental factor has higher explanatory power, which may be because of the relatively dispersed distribution of flow-like landslides. Regarding the spatial distribution of flow-like landslides, lithology is the only factor that differs from the other environmental factors.

The top 3 dominant interactions for flow-like landslides are the interaction between lithology and elevation (0.1292), the interaction between lithology and AMT (0.1269) and the interaction between lithology and AMP (0.1047). Lithology is also a very important factor in the interactions because detrital material produced by rock weathering is the main source of flow-like landslides, and the physical properties of rocks are closely related to the occurrence of complex landslides.

From these results, it can be concluded that: ① all the dominant interactions are nonlinearly enhanced, which means that the *q* value of the interaction of two factors is higher than the sum of the *q* values of the individual. ② Although the dominant interaction types and *q*-values varied with respect to different types of landslides, interactions between environmental factors significantly enhance the spatial distribution of landslides.

**Identification of high-risk areas of landslides contrasted with the results of the CF model.** Identifying high-risk areas is critical for understanding the development of regional landslides [63]. In the risk detector of GeoDetector, a *t*-test is used to compare the differences in average values between strata [23]; the larger the difference is, the larger the danger of landslides in the subregion. The CF model recognizes the high-risk zones by computing the landslide frequency in different areas [59–61]. The results of the risk detector of the GeoDetector (GD) and CF models (Table 6 and Fig 10) are calculated, the similarities and differences between them are analyzed, and the high-risk zones in our study area are achieved based on these results.

**Table 6. The CF values and risk detector results of geodetector calculated between landslides and environmental factors.**

| Factors | Class | No. | CF-rockslide | CF-rock fall | CF-flow-like | CF-total landslide | GD-rockslide | GD-rock fall | GD-flow-like | GD-total landslide |
|---|---|---|---|---|---|---|---|---|---|---|
| Fault distance (m) | 0–500 | 1 | 0.323 | 0.331 | 0.120 | 0.26 | 0.067 | 0.034 | 0.051 | 0.150 |
| | 500–1000 | 2 | 0.218 | 0.277 | 0.047 | 0.17 | 0.060 | 0.031 | 0.052 | 0.142 |
| | 1000–1500 | 3 | 0.059 | 0.065 | 0.036 | 0.05 | 0.050 | 0.028 | 0.048 | 0.127 |
| | 1500–2000 | 4 | -0.185 | -0.014 | 0.086 | -0.04 | 0.042 | 0.020 | 0.050 | 0.113 |
| | 2000–2500 | 5 | -0.226 | -0.475 | -0.073 | -0.21 | 0.036 | 0.021 | 0.049 | 0.109 |
| | >2500 | 6 | -0.434 | -0.535 | -0.135 | -0.32 | 0.024 | 0.009 | 0.038 | 0.072 |
| Lithology | L1 | 7 | -0.155 | -1.000 | -0.185 | -0.33 | 0.034 | 0.001 | 0.032 | 0.070 |
| | L2 | 8 | -0.754 | -1.000 | 0.746 | 0.42 | 0.026 | 0.002 | 0.156 | 0.182 |
| | L3 | 9 | -0.285 | -0.488 | 0.225 | -0.05 | 0.029 | 0.010 | 0.067 | 0.104 |
| | L4 | 10 | -0.596 | -1.000 | -0.225 | -0.52 | 0.017 | 0.000 | 0.045 | 0.065 |
| | L5 | 11 | 0.275 | -0.869 | -0.752 | -0.32 | 0.070 | 0.006 | 0.021 | 0.096 |
| | L6 | 12 | 0.330 | -0.146 | 0.110 | 0.18 | 0.069 | 0.017 | 0.052 | 0.137 |
| | L7 | 13 | 0.351 | 0.712 | -0.504 | 0.35 | 0.073 | 0.083 | 0.022 | 0.175 |
| | L8 | 14 | 0.106 | -0.842 | 0.114 | -0.05 | 0.050 | 0.005 | 0.051 | 0.106 |
| | L9 | 15 | -0.012 | 0.130 | -0.120 | -0.01 | 0.045 | 0.030 | 0.038 | 0.113 |
| Epicenter distance (km) | 0–6 | 16 | 0.156 | 0.323 | -0.030 | 0.14 | 0.058 | 0.042 | 0.045 | 0.145 |
| | 6–12 | 17 | -0.048 | -0.053 | -0.067 | -0.06 | 0.044 | 0.020 | 0.042 | 0.106 |
| | 12–18 | 18 | -0.109 | -0.567 | -0.018 | -0.15 | 0.038 | 0.010 | 0.047 | 0.094 |
| | 18–25 | 19 | -0.130 | -0.574 | 0.201 | -0.05 | 0.037 | 0.008 | 0.062 | 0.106 |
| | 25–30 | 20 | -0.186 | -0.418 | 0.268 | 0.00 | 0.035 | 0.012 | 0.057 | 0.103 |
| | 30–37 | 21 | -0.756 | -0.834 | 0.074 | -0.42 | 0.017 | 0.003 | 0.041 | 0.061 |
| | >37 | 22 | 0.412 | -1.000 | -1.000 | -0.33 | 0.048 | 0.000 | 0.007 | 0.045 |
| Elevation (m) | <1500 | 23 | 0.380 | 0.528 | -0.692 | 0.16 | 0.099 | 0.075 | 0.032 | 0.199 |
| | 1500–2000 | 24 | 0.429 | 0.812 | -0.549 | 0.51 | 0.082 | 0.107 | 0.022 | 0.205 |
| | 2000–2500 | 25 | 0.418 | 0.730 | -0.517 | 0.41 | 0.101 | 0.093 | 0.033 | 0.220 |
| | 2500–3000 | 26 | 0.377 | 0.368 | -0.553 | 0.14 | 0.067 | 0.037 | 0.039 | 0.142 |
| | 3000–3500 | 27 | 0.059 | -0.153 | -0.222 | -0.09 | 0.052 | 0.023 | 0.045 | 0.121 |
| | 3500–4000 | 28 | -0.094 | -0.427 | 0.054 | -0.09 | 0.035 | 0.011 | 0.051 | 0.097 |
| | 4000–4500 | 29 | -0.391 | -0.800 | 0.247 | -0.12 | 0.030 | 0.006 | 0.052 | 0.087 |
| | 4500–5000 | 30 | -0.283 | -0.907 | 0.320 | -0.05 | 0.025 | 0.002 | 0.058 | 0.085 |
| | 5000–5500 | 31 | -0.545 | -1.000 | -0.102 | -0.45 | 0.022 | 0.001 | 0.043 | 0.067 |
| | >5500 | 32 | -0.754 | -1.000 | -0.881 | -0.85 | 0.008 | 0.000 | 0.020 | 0.030 |
| Slope (°) | <10 | 33 | -0.834 | -0.924 | -0.770 | -0.82 | 0.036 | 0.009 | 0.034 | 0.080 |
| | 10–19 | 34 | -0.592 | -0.830 | -0.361 | -0.54 | 0.038 | 0.014 | 0.049 | 0.101 |
| | 19–27 | 35 | -0.266 | -0.659 | -0.104 | -0.27 | 0.040 | 0.015 | 0.043 | 0.098 |
| | 27–33 | 36 | 0.029 | -0.324 | 0.005 | -0.05 | 0.041 | 0.022 | 0.050 | 0.112 |
| | 33–39 | 37 | 0.169 | -0.071 | 0.201 | 0.15 | 0.049 | 0.024 | 0.049 | 0.122 |
| | 39–44 | 38 | 0.269 | 0.380 | 0.212 | 0.27 | 0.051 | 0.027 | 0.044 | 0.122 |
| | 44–50 | 39 | 0.315 | 0.455 | 0.194 | 0.31 | 0.056 | 0.037 | 0.052 | 0.143 |
| | 50–59 | 40 | 0.299 | 0.615 | 0.085 | 0.35 | 0.053 | 0.042 | 0.049 | 0.144 |
| | 59–88 | 41 | 0.274 | 0.798 | -0.056 | 0.48 | 0.046 | 0.025 | 0.054 | 0.124 |

*(Continued)*

**Table 6.** (Continued)

| Factors | Class | No. | CF-rockslide | CF-rock fall | CF-flow-like | CF-total landslide | GD-rockslide | GD-rock fall | GD-flow-like | GD-total landslide |
|---|---|---|---|---|---|---|---|---|---|---|
| Aspect | Flat | 42 | -1.000 | -1.000 | -1.000 | -1.00 | 0.037 | 0.003 | 0.011 | 0.057 |
| | North | 43 | -0.389 | -0.858 | -0.369 | -0.47 | 0.036 | 0.021 | 0.043 | 0.102 |
| | Northeast | 44 | -0.164 | -0.743 | -0.111 | -0.25 | 0.040 | 0.019 | 0.050 | 0.109 |
| | East | 45 | -0.083 | -0.069 | -0.167 | -0.11 | 0.050 | 0.020 | 0.046 | 0.117 |
| | Southeast | 46 | 0.208 | 0.470 | 0.079 | 0.25 | 0.051 | 0.027 | 0.049 | 0.125 |
| | South | 47 | 0.293 | 0.532 | 0.377 | 0.39 | 0.047 | 0.024 | 0.053 | 0.121 |
| | Southwest | 48 | 0.339 | 0.253 | 0.346 | 0.33 | 0.055 | 0.027 | 0.055 | 0.135 |
| | West | 49 | -0.166 | -0.652 | -0.183 | -0.26 | 0.045 | 0.027 | 0.045 | 0.116 |
| | Northwest | 50 | -0.548 | -0.918 | -0.576 | -0.63 | 0.039 | 0.017 | 0.036 | 0.093 |
| | North | 51 | -0.588 | -0.800 | -0.603 | -0.63 | 0.033 | 0.019 | 0.036 | 0.089 |
| River density | Sparse | 52 | -0.578 | -0.942 | -0.247 | -0.50 | 0.021 | 0.003 | 0.037 | 0.063 |
| | Relatively sparse | 53 | -0.224 | -0.493 | 0.181 | -0.05 | 0.035 | 0.014 | 0.056 | 0.105 |
| | Relatively dense | 54 | 0.175 | 0.320 | 0.035 | 0.17 | 0.058 | 0.038 | 0.052 | 0.147 |
| | Dense | 55 | 0.476 | 0.505 | -0.453 | 0.31 | 0.088 | 0.047 | 0.029 | 0.161 |
| River distance (m) | 0–500 | 56 | -0.283 | 0.286 | -0.581 | -0.22 | 0.065 | 0.034 | 0.043 | 0.142 |
| | 500–1000 | 57 | 0.364 | 0.315 | -0.154 | 0.22 | 0.053 | 0.031 | 0.053 | 0.136 |
| | 1000–1500 | 58 | 0.234 | -0.133 | 0.322 | 0.23 | 0.042 | 0.028 | 0.048 | 0.114 |
| | 1500–2000 | 59 | -0.083 | -0.349 | 0.466 | 0.23 | 0.039 | 0.020 | 0.049 | 0.105 |
| | 2000–2500 | 60 | -0.453 | -0.716 | 0.332 | -0.09 | 0.033 | 0.021 | 0.058 | 0.098 |
| | >2500 | 61 | -0.525 | -0.945 | -0.287 | -0.50 | 0.021 | 0.009 | 0.037 | 0.063 |
| AMP (mm) | 600–700 | 62 | 0.597 | 0.727 | -0.590 | 0.50 | 0.112 | 0.099 | 0.032 | 0.236 |
| | 700–800 | 63 | 0.297 | 0.417 | -0.228 | 0.21 | 0.074 | 0.043 | 0.043 | 0.157 |
| | 800–900 | 64 | -0.146 | -0.564 | 0.132 | -0.07 | 0.038 | 0.013 | 0.055 | 0.107 |
| | 900–1000 | 65 | -0.520 | -0.890 | 0.086 | -0.28 | 0.021 | 0.003 | 0.047 | 0.073 |
| | 1000–1300 | 66 | -0.629 | -1.000 | -0.412 | -0.61 | 0.011 | 0.002 | 0.024 | 0.038 |
| AMT (°) | <0 | 67 | -0.483 | -1.000 | 0.077 | -0.32 | 0.022 | 0.002 | 0.046 | 0.071 |
| | 0–2 | 68 | -0.468 | -0.896 | 0.231 | -0.19 | 0.027 | 0.003 | 0.056 | 0.086 |
| | 2–5 | 69 | -0.357 | -0.657 | 0.175 | -0.15 | 0.031 | 0.008 | 0.053 | 0.093 |
| | 5–8 | 70 | 0.116 | -0.238 | -0.176 | -0.06 | 0.047 | 0.019 | 0.044 | 0.112 |
| | 8–12 | 71 | 0.340 | 0.400 | -0.303 | 0.20 | 0.075 | 0.046 | 0.041 | 0.160 |
| | >12 | 72 | 0.497 | 0.721 | -0.484 | 0.46 | 0.094 | 0.095 | 0.029 | 0.211 |
| VFC | Unveg | 73 | -0.389 | -0.741 | -0.097 | -0.31 | 0.034 | 0.016 | 0.043 | 0.095 |
| | Low | 74 | -0.134 | 0.232 | 0.202 | 0.11 | 0.040 | 0.019 | 0.045 | 0.105 |
| | Medium | 75 | 0.123 | 0.404 | 0.120 | 0.20 | 0.048 | 0.018 | 0.054 | 0.119 |
| | High | 76 | 0.323 | 0.187 | -0.078 | 0.18 | 0.066 | 0.039 | 0.047 | 0.148 |
| Land cover | Cultivated land | 77 | -0.156 | -0.570 | -1.000 | -0.58 | 0.077 | 0.012 | 0.027 | 0.116 |
| | Forest | 78 | 0.061 | -0.287 | -0.311 | -0.12 | 0.056 | 0.035 | 0.043 | 0.132 |
| | Shrub land | 79 | 0.199 | 0.514 | 0.235 | 0.32 | 0.045 | 0.020 | 0.050 | 0.115 |
| | Grassland | 80 | 0.640 | 0.821 | 0.700 | 0.72 | 0.034 | 0.006 | 0.069 | 0.100 |
| | wetland | 81 | -1.000 | -1.000 | -1.000 | -1.00 | 0.000 | 0.000 | 0.048 | 0.045 |
| | waterbodies | 82 | -1.000 | -1.000 | -1.000 | -1.00 | 0.063 | 0.023 | 0.018 | 0.102 |
| | Artificial surfaces | 83 | -1.000 | -1.000 | -1.000 | -1.00 | 0.024 | 0.007 | 0.041 | 0.078 |
| | Perennial snow or ice | 84 | -0.403 | -0.796 | 0.209 | -0.14 | 0.026 | 0.004 | 0.051 | 0.083 |

Note: For the class of lithology, L1 represents modern glacier, L2 is Middle Permian limestone and sandy slate, L3 is Miocene granite, L4 is Jurassic carbonate rock with clastic rock interlayers, L5 is Quaternary deposits, L6 is Early Carboniferous slate and phyllite rock, L7 is Late Jurassic mélange, L8 is Jurassic flysch clastic rock, and L9 is Presinian deep metamorphic rock.

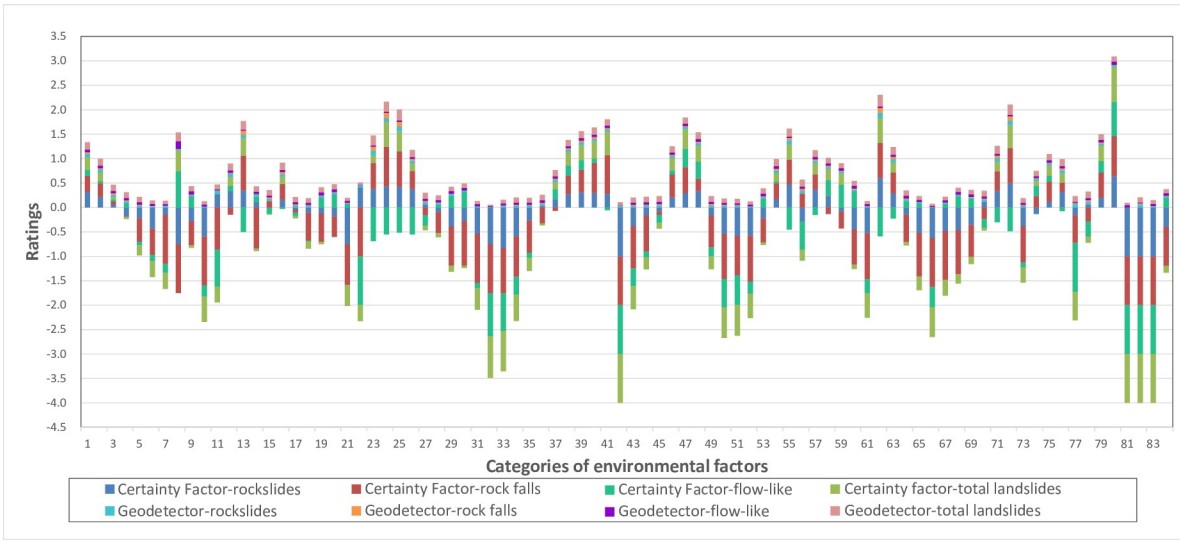

**Fig 10. Graphical representation of the CF value and risk detector result of GeoDetector.** The number on the horizontal axis has the same meaning as the number listed in Table 6.

*(1) Fault, lithology and epicenter.* The risk detector shows that the closer an area is to a fault, the higher the probability of landslide occurrence. The zones with 0–500 m distances from faults are the highest risk areas for all types of landslides, except for the flow-like landslides, the highest value of which is 500–1000 m away from faults, which is still very close to faults. The correlation between the distribution of landslides and fault distance given by the CF model is similar. The 0–500 m distance away from the fault is the high-risk area for all types of landslides, and the landslide probability decreases as the fault distance increases, except for flow-like landslides, which have the highest CF value in the zone of 1500–2000 m of distance.

In the case of lithology, the results of the GeoDetector risk detector show that the lithology of the middle Permian limestone sandy slate is most prone to landslides. For the 3 types of landslides, the Late Jurassic mélange is the most susceptible to rockslides and rock falls, and the lithology of the middle Permian limestone and sandy slate is the most prone to danger in terms of flow-like landslides. The results of the CF model are the same as the results of the risk detector. The Late Jurassic mélange is the most prone to rockslides and rock falls, and middle Permian limestone and sandy slate are the most prone to flow-like landslides. Previous studies [3, 5] have confirmed that the interaction between soft rock and hard rock is one of the conditions for the occurrence of landslides.

Similar to the fault distance, the risk detector results of the epicenter distance show that the values closest to the epicenter (0–6 km) are highest for all types of landslides except for flow-like landslides; however, they do not strictly follow the pattern of decreasing landslide probability with increasing distance. The CF value of the epicenter of 0–6 km is the highest and only has a positive CF value for landslides, and the CF values also do not strictly follow the pattern of decreasing landslide probability with increasing distance for the 3 types of landslides as the risk detector results.

*(2) Elevation, slope, and aspect.* The risk detector results of elevation vary with the type of landslide. The highest value for rockslides drops into the range of 2500 to 3000 m. Furthermore, the probability of rockslide occurrence decreases as the altitude increases. For rock falls, the highest value ranges from 1500 to 2000 m, and then the risk detector value decreases to 0 when the altitude is higher than 5500 m, which means no rock fall is found at that altitude.

However, the highest value for flow-like landslides ranges from 4500 to 5000 m, and the risk detector value decreases with both increasing and decreasing elevation ranges. The results of the CF model of elevation are similar to the results of the risk detector. For rockslides and rock falls, the highest value drops into the range of 1500 to 2000 m, the value decreases as the altitude increases, and the highest value for flow-like landslides drops into the range of 4500 to 5000 m.

The highest value for rockslides occurred in slope classes of 44˚-50˚. For rock falls, the highest value drops into the slope class of 50˚-59˚ and 59˚-88˚ for flow-like landslides. The results indicate that landslide-prone terrain is relatively steep (>44˚) in the study area. The CF values of the slope also show that landslide-prone terrain is relatively steep (>44˚) in the study area except for flow-like landslides, in which the highest value of CF drops into the range of 39˚-44˚.

In the case of the aspect, south- and southwest-facing slopes are the most prone to landslides, as they have the highest risk detector value. The CF values of the south- and southwest-facing slopes are also the highest, which means that south- and southwest-facing slopes are the most prone to landslides. These aspects are the sunny sides, receiving the most intense light, which can accelerate rock weathering and form landslides [64].

*(3) River.* The erosion of rivers will cause slope instability [6]. The value of the risk detector of river density shows that the denser the rivers are, the more prone they are to landslides, except for flow-like landslides, which have the highest value in relatively sparse areas of river density. In the case of river distance, the risk detector value decreases as river distance increases, which means that the closer to the river, the higher the occurrence probability of the landslides, except for flow-like landslides, which have the highest value occurring in the range of 2000 to 2500 m away from the river. The CF values of river density are similar to the values of the risk detector; the denser the rivers are, the more prone they are to landslides, except for flow-like landslides, which have the highest value in relativity dense areas of river density. The CF value of river distance shows that 500–1000 m away from rivers is the highest occurrence range, which is different from the results of the risk detector in that the range of 0–500 m away from rivers is most prone to landslides.

*(4) AMP and AMT.* According to the results of the risk detector, the AMP and AMT ranges defining the high-risk areas are 600–700 mm and 12–20˚C, respectively, except for flow-like landslides, which are 800–900 mm and 0–2˚C, respectively. The CF value results are the same as the risk detector results. Contrary to the conventional concept, the areas prone to landslides are not those with the largest AMP, which may be because landslides are more susceptible to instantaneous rainfall rather than the average annual rainfall.

*(5) Land cover and VFC.* Changes in land use and land cover can change landslide susceptibility in mountainous areas [65]. According to the VFC results, high VFC areas rather than low VFC areas are prone to landslides, except for flow-like landslides, and medium VFC areas are the areas most at risk. Regarding the risk detector value of land cover, the situation changes with the type of landslide; cultivated land is most prone to rockslides, forest is most prone to rock falls, and grassland is most prone to flow-like landslides. The CF values of the VFC are different with different types of landslides; the highest value for rockslides has high vegetation coverage, the highest value for rock falls has medium vegetation coverage, and the highest value for flow-like landslides has low vegetation coverage. Regarding the CF value of land cover, grassland is most susceptible to all types of landslides, which are different from the results of the risk detector.

The results of the high-risk areas show that the GeoDetector and CF models are similar with most environmental factors in high-risk area detection and in the tendency of variation of landslide spatial distribution; the influence of environmental factors on flow-like landslides

is significantly different from that of rockslides and rock falls and may be caused by the complex constitution of flow-like landslides, including debris flows, mudflows and complex landslides, which increases the complexity of their spatial distribution.

Combining the calculated results of risk zones and dominant single and interaction factors, the important interaction for rockslides is between lithology and AMP ($q = 0.2201$), the Late Jurassic mélange areas with annual mean precipitation between 600–700 mm are prone to rockslides, the important interaction for rock falls is between epicenter distance and elevation ($q = 0.2201$), the areas 0–6 km away from the epicenter and elevation range of 2500–3000 m are prone to rock falls, the important interaction for flow-like landslides is between lithology and elevation (q = 0.1292), and the middle Permian limestone, sandy slate and elevation range of 4500–5000 m are prone to flow-like landslides.

## Conclusion

The study area is located in the southeastern Tibetan Plateau, China, where the geological and geographical environment is unique, landslides occur frequently, and the area acts as the key access into Tibet. Since landslides pose a great threat to life, property and transportation, their distribution and spatial relationship with environmental factors are extremely important for future planning and disaster management. For this purpose, we improved the landslide inventory and acquired 1766 landslides based on the comprehensive interpretation of multiple satellite images and DEM data. Three types of landslides were identified based on their geometric and spectral features, including 698 rockslides, 343 rock falls, and 725 flow-like landslides. The effects of environmental factors on landslides in the study area are investigated using geographical detector and certainty factor methods.

Geographical detector analysis reveals the explanatory power of individual environmental factors and interactions. For different types of landslides, AMP is the dominant factor for rockslides, elevation is the dominant factor for rock falls and lithology is the dominant factor for flow-like landslides. The results of the interaction detector showed that the interactions among environmental factors would significantly enhance the impacts on landslides. The top 3 dominant interactions for all types of landslides were revealed. The risk detectors of the GeoDetector and CF models were used to identify the high-risk areas of landslides by rating the subregions of the environmental factors. Similar results reflected the credibility of the high-risk areas in identifying results.

Our method offers a quantitative and objective analytical method that can be used to reveal the spatial association between landslides and environmental factors for the prevention and mitigation of geological disasters. Although our research has yielded certain results, there are also some shortcomings. Due to the limitations of resolution and timeliness of remote sensing images, only landslides over a certain scale might be identified, which may result in an insufficient number of regional samples. When continuous data are converted to categorical data, the singleness of the adopted methods may affect the analysis results. Therefore, the influences of insufficient samples and diverse discrete methods should be considered in future studies.

## Acknowledgments

We would like to express our gratitude to Prof. Wang zhihua and Prof. Zhao Zhifang for constructive comments and their help on our thesis.

## Author Contributions

**Conceptualization:** Wei-Jie Jia, Cheng-Hu Zhou.

**Formal analysis:** Wei-Jie Jia, Meng-Fei Wang.

**Investigation:** Meng-Fei Wang, Qing-Hua Yang.

**Methodology:** Wei-Jie Jia.

**Writing – original draft:** Wei-Jie Jia.

**Writing – review & editing:** Cheng-Hu Zhou, Qing-Hua Yang.

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
