## [Decision Letter · Decision Letter 0]

10 Jun 2020

PONE-D-20-11323

Spatial association analysis between landslides and environmental factors based on geographical detector in southeastern of Tibet Plateau, China

PLOS ONE

Dear Dr. jia,

Thank you for submitting your manuscript to PLOS ONE. After careful consideration, we feel that it has merit but does not fully meet PLOS ONE’s publication criteria as it currently stands. Therefore, we invite you to submit a revised version of the manuscript that addresses the points raised during the review process.

We look forward to receiving your revised manuscript.

Kind regards,

Shamsuddin Shahid

Academic Editor

PLOS ONE

Additional Editor Comments:

All the reviewers believe the article is interesting. However, they asked for deeper review of literature, improvement of English language and quality of figures. I completely agree with all the comments of the reviewers. Particularly, I believe literature review is really very shallow. Numerous studies have been conducted to understand the environmental factors responsible for landslides. Many of them published in PLOS ONE. Authors need to provide a deeper literature review and establish the novelty of the study clearly.

3. We note that Figures 1-7 in your submission contain map/satellite images which may be copyrighted.

a. You may seek permission from the original copyright holder of Figures 1-7 to publish the content specifically under the CC BY 4.0 license. 

4. Please clarify whether all the data is available. In your Data availability statement, please provide references and links or contact information to all the datasets used. Was any data collected specifically for this study in the field, and if so, were any field permits obtained?

Reviewers' comments:

Reviewer's Responses to Questions

**Comments to the Author**

1. Is the manuscript technically sound, and do the data support the conclusions?

Reviewer #1: Yes

Reviewer #2: Yes

Reviewer #3: Yes

2. Has the statistical analysis been performed appropriately and rigorously? 

Reviewer #1: Yes

Reviewer #2: Yes

Reviewer #3: Yes

3. Have the authors made all data underlying the findings in their manuscript fully available?

Reviewer #1: Yes

Reviewer #2: Yes

Reviewer #3: Yes

4. Is the manuscript presented in an intelligible fashion and written in standard English?

Reviewer #1: No

Reviewer #2: Yes

Reviewer #3: Yes

5. Review Comments to the Author

Reviewer #1: Comments for Authors

The manuscript “Spatial association analysis between landslides and environmental factors based on geographical detector in southeastern of Tibet Plateau, China” present an interesting study on detecting landslides, and their prevention or mitigation. The manuscript is well structured and nicely presented. However, I have some observations and comments for the authors as follows.

Topic

Authors should remove ‘of’ in the topic or put include ‘part’ before ‘of’

Introduction

Line 51 – 53: Change ‘resulting’ to ‘resulted’

Line 53: remove ‘in the’ before upstream

Line 58 – 60: One inverted comma is after China, where is the one before?

Line 70 – 75: Through data …… Sentence should be split into two.

There are other grammatical errors in this section, authors should please check and make corrections.

Study area and data

Line 120: change ‘blocked’ to ‘trapped’ and change ‘condensation’ to ‘condenses’. Please check and use appropriate words and check other grammatical errors.

Line 131: change ‘immigrated’ to ‘brought’.

Is there any map of the study area? I suggest authors include one.

Change ‘geology environment’ to ‘geological environment’.

Landslide inventory

Line 166 - 167: change ‘combining’ to ‘combined’ and change ‘identify’ to ‘identification’.

Line 172 – 174: Is this sentence not the same/almost the same as the one in line 153 – 155? If it is, then it should be deleted or the earlier one can be referred to if need be.

Line 184: change ‘main’ to ‘mainly’. Please check other typographical and grammatical errors.

Line 185: remove ‘et al’ and replace with ‘and so on’.

Line 199: what is been referred to as book here? If authors mean the ‘inventory’, this should be changed to ‘record’.

Environmental factors

Line 207: replace ‘predecessors’ with another word.

Line 215 – 216: authors cite some of the papers that have done these verifications.

Line 217: change ‘adapted’ to ‘considered’.

Table 2. shows there are 13 factors. Differentiate the Y variable from the 12 factors in the text.

Line 238: change ‘import’ to ‘important’.

Line 250: change ‘of our study area’ to ‘in our study area’

Line 253: change ‘show’ to ‘shown’.

Method

GeoDetector

Line 328: remove ‘other’

Results and Discussion

Landslides inventory and their spatial distribution

Line 354 – 356: authors should mention the reasons for the heterogeneity of the landslides in the area?

Line 356 – 361: why do these areas have higher concentrations of landslides?

Line 363 – 365: why is there more rock falls in the area? Are these due to the elevation/slope or the geology of the area?

Line 369 – 371: Authors should also mention the likely factors for these types of landslides in the area since there are different factors such as type of overburden/soil, slope and climatic factors.

Quantitative analysis of spatial association between the geological disasters with the environmental factors

Line 376 – 377: change heading to “Quantitative analysis of spatial association between geological disasters and environmental factors”

Line 378: This heading is not clear. Do the authors mean ‘significance and interaction of environmental factors?’

Line 392: put ‘of’ after because

Line 395: change ‘respecting’ to ‘with respect to’

Line 400: change ‘respecting’ to ‘with respect to’ in table. Also change other ones in the manuscript.

Line 415 - 423: authors should use comma at the appropriate places instead of ‘and’.

Identification of high risk areas of landslides and contrast between CF model and GeoDetector

Line 450: is this supposed to be expect or except?

Line 459 – 460: authors should give the possible reasons for the decrease in the probability of occurrence of landslide with altitude. Authors should mention why the risk detector value decreased with altitude higher than 5500m for collapse also.

Line 470: change ‘expect’ or ‘except’.

Line 487: change ‘same as’ to ‘similar to’.

Line 505 - 508: what could be the causes in these variations? Vegetation cover, depending on types has been suggested to prevent landslide. See following studies:

Kobayashi, Y., Mori, A.S. The Potential Role of Tree Diversity in Reducing Shallow Landslide Risk. Environmental Management 59, 807–815 (2017). https://doi.org/10.1007/s00267-017-0820-9

Mao et al (2012) Engineering ecological protection against landslides in diverse mountain forests: Choosing cohesion models. https://doi.org/10.1016/j.ecoleng.2011.03.026

Line 553: use another word for ‘shortages’ e.g. use shortcomings.

Other comments

Authors should improve the texts in some of the figures that are not legible.

There are a lot grammatical and typographical errors in the manuscript. Authors should seek the help of a native or better English speaker to help improve the grammar of the manuscript.

Reviewer #2: The main objective of manuscript is to explore the spatial association between the landslides and the environmental factors in the southeastern part of Tibet Plateau. The subject of study is interesting and article is generally well structured. However, I would like to recommend the authors to improve the paper considering the following comments.

1-Many studies have been conducted to analysis influencing factors on landslides as a typical disaster in china. Author(s) also mentioned some of them in introduction section. The concluding remark of literature review in Introduction is “none of these studies would highlight the dominant factors or the interaction between the factors, and the existing landslides investigation is incomplete”. However, the new findings of this study should be clear and must compare to other studies in discussion and authors should emphasize on their new finding. Manuscript needs more about innovation and validation.

2-Geographical detector tool has applied to detect the dominant factors; this statistical method includes four parts which are risk detector, factor detector, ecological detector, and interaction detector. Author(s) are recommended to provide some information if other sophisticated methods are available to do such analysis and provide evidence for the significance and preference of their methodology based on other studies.

3-It is clear that different factors and multi-factor interaction can influence landslides, but the challenge is to find what those interactions are, how much influence of which factor, how the changes on those factors influence the changes of landslides and what can be the reasons of them. It is required to put more scientific interpretation of results and I suggest the authors to improve the discussion.

4- The quality of the figures is very poor. Please replace all with good ones. The figure1 were not defined in terms of A, B, C, D, F. Landslide inventory map can provide with good quality and the location of study area can be presented.

5- Grammar and English writing need to improve.

6-keywords should insert after abstract and before introduction.

7-For the link: http://www. GeoDetector.org, Please check as it does not exist, do you mean http://www.geodetector.cn/ ?

8- You May reconsider about your title as there is much similarity with below reference:

Yue, X. L., Wu, S. H., Huang, M., Gao, J. B., Yin, Y. H., Feng, A. Q., & Gu, X. P. (2018). Spatial association between landslides and environmental factors over Guizhou Karst Plateau, China. Journal of Mountain Science, 15(9), 1987-2000. https://link.springer.com/article/10.1007%2Fs11629-018-4909-2

Reviewer #3: The authors of the manuscript “Spatial association analysis between landslides and environmental factors based on geographical detector in southeastern of Tibet Plateau, China” tried to investigate the number of landslides and their relationships with certain factors in southeastern of Tibet Plateau. They used geographical detector method and confirm its suitability with Certainty factor model. The study considered and remote sensing data obtained from different sources to conduct the study. Authors claimed that previous studies over the study area were limited in terms of data and mostly conducted over areas which were accessible. I explored the literature review and found that a large body of literature is available where studies are conducted similar to the objectives of present study over the study area. I believe authors should conduct a deeper literature review to highlight the gaps from previous studies. Almost all results of the study are based on remote sensing data, validation of the results of the study is important. Is there any possibility to validate the results of the study?. The spatial patterns of precipitation and temperature in figure (figure numbers are not given) looks different, please carefully check them. The quality of images are very poor, they must be improved as study is conducted based on those images. Figures should be properly numbers; it’s difficult to understand which figure are authors referring. The results of correlating factors are not clear. Authors should clearly provide the spatial correlation of landslides with environmental factors.

6. PLOS authors have the option to publish the peer review history of their article (what does this mean?). If published, this will include your full peer review and any attached files.

Reviewer #1: No

Reviewer #2: No

Reviewer #3: No

---

## [Author Response · Author response to Decision Letter 0]

11 Feb 2021

Dear Dr. Shahid:

Thank you for giving us the opportunity to summit a revised draft of the manuscript “Spatial association analysis between landslides and environmental factors based on geographical detector in southeastern of Tibet Plateau, China” for publication in the PLOS ONE. We appreciate the valuable time and effort that you and reviewers dedicated to providing feedback on our manuscript. Your comments are insightful and valuable, which make us rethink and improve the paper. We are incorporated most of the suggestions made by the reviewers. The finished draft is revised by American Journal Experts (AJE) on language editing and manuscript formatting to ensure our manuscript meets your submission guideline. Those changes are provided with name “Revised Manuscript with Track Changes”. The point-by-point response to the reviewers’ comments and concerns in blue, please see below. All page numbers in the author response refer to the revised manuscript file without tracked changes.

Reviewer note: 

The main objective of manuscript is to explore the spatial association between the landslides and the environmental factors in the southeastern part of Tibet Plateau. The subject of study is interesting and article is generally well structured.

Author response: Thank you!

1. Many studies have been conducted to analysis influencing factors on landslides as a typical disaster in China. Author(s) also mentioned some of them in introduction section. The concluding remark of literature review in Introduction is “none of these studies would highlight the dominant factors or the interaction between the factors, and the existing landslides investigation is incomplete”. However, the new findings of this study should be clear and must compare to other studies in discussion and authors should emphasize on their new finding. Manuscript needs more about innovation and validation.

Author response: Thank you for pointing this out, the reviewer is correct, the former concluding remark of literature review “none of these studies would highlight the dominant factors or the interaction between the factors, and the existing landslides investigation is incomplete” in Introduction is inappropriate, therefore the literature review (Line56-line112) on introduction is rewritten, we have add many new references of relevant literatures, the number of references increased from 42 to 65, compared most frequently used relationship analyses methods between environmental factors and landslides. The studies on southeastern Tibet also mentioned on Line79 to line97. Because the complexity relationship between landslides and environmental factors especially on complex terrain regions as our study area, our purpose is to find an explicit and systematic explanation for landslides prevention and mitigation. And the new method and targets of this paper are revealed on line98 to line112.

2. Geographical detector tool has applied to detect the dominant factors; this statistical method includes four parts which are risk detector, factor detector, ecological detector, and interaction detector. Author(s) are recommended to provide some information if other sophisticated methods are available to do such analysis and provide evidence for the significance and preference of their methodology based on other studies.

Author response: As suggested by the reviewer, on literature review, we analyses most frequently used methods, which include deterministic approaches, statistical analyses, and computational intelligence methods, in detail, and their significance and preference (line64-line78). But because of the complexity of the relationship between landslides and environmental factors, explicit and systematic explanations are rare. 

3. It is clear that different factors and multi-factor interaction can influence landslides, but the challenge is to find what those interactions are, how much influence of which factor, how the changes on those factors influence the changes of landslides and what can be the reasons of them. It is required to put more scientific interpretation of results and I suggest the authors to improve the discussion.

Author response: Thank you for point this out, we think this is an excellent suggestion. We have improved our discussion on chapter of “Results and discussion”. In part of “The dominant factors and interactions of the environmental factors” (line406-line488), we analyses the dominant factors and interaction of multi-factor of total landslides and each type of landslides. We explained how much influence of each factor by the value of q, which means explanatory power in geographical detector. How the interaction of environmental factors enhances the influence on landslides also revealed based on interaction detector. We compared our results with previous study and other relevant evidence, and tried to explain the reasons for each of our results. 

4. The quality of the figures is very poor. Please replace all with good ones. The figure1 were not defined in terms of A, B, C, D, F. Landslide inventory map can provide with good quality and the location of study area can be presented.

Author response: As suggested by the reviewer, we improved the quality of each figure, and replace all with good one. We newly increased “Figure 1. Map of the Study Area” (line149) to present the location of study area. And the new figure2 is defined in terms of A, B, C, D, F. Figure4,5,6,7 is all modified according to the rules. 

5. Grammar and English writing need to improve.

Author response: Thank you for point this out, for improve grammar and English writing, we chose the professional scientific editing service of “American Journal Experts (AJE) ” for language editing and manuscript formatting to ensure our manuscript meets your submission guideline. 

6. keywords should insert after abstract and before introduction.

Author response: While we appreciate the reviewer’s feedback, after check the formatting documents of “PLOSOne_formatting_sample_main_body.pdf”, and the comments of Academic formatting specialist of AJE, the journal does not require the inclusion of this information in our manuscript, so we delete the keywords which insert before. The track can check in the document of “Revised Manuscript with Track Changes”.

7. For the link: http://www. GeoDetector.org, Please check as it does not exist, do you mean

http://www.geodetector.cn/ ?

Author response: Thank you for point this out, after check the link again, we change it to http://www.geodetector.cn/(line309).

8. You May reconsider about your title as there is much similarity with below reference:

Yue, X. L., Wu, S. H., Huang, M., Gao, J. B., Yin, Y. H., Feng, A. Q., & Gu, X. P. (2018). Spatial

association between landslides and environmental factors over Guizhou Karst Plateau, China. Journal of Mountain Science, 15(9), 1987-2000. https: // link. springer. Com / article / 10. 1007% 2Fs11629-018-4909-2.

Author response: Thank you for point this out, after compare the significance and preference of these reference, we change our title as “Analysis of the spatial association of geographical detector-based landslides and environmental factors in the southeastern Tibetan Plateau, China” .

---

## [Decision Letter · Decision Letter 1]

26 Mar 2021

PONE-D-20-11323R1

Analysis of the spatial association of geographical detector-based landslides and environmental factors in the southeastern Tibetan Plateau, China

PLOS ONE

Dear Dr. jia,

Thank you for submitting your manuscript to PLOS ONE. After careful consideration, we feel that it has merit but does not fully meet PLOS ONE’s publication criteria as it currently stands. Therefore, we invite you to submit a revised version of the manuscript that addresses the points raised during the review process.

We look forward to receiving your revised manuscript.

Kind regards,

Shamsuddin Shahid

Academic Editor

PLOS ONE

Journal Requirements:

2) Please state in your Methods section whether any permits were obtained for the field work, or if no field permits were obtained, a short statement as to why not.

Reviewers' comments:

Reviewer's Responses to Questions

**Comments to the Author**

1. If the authors have adequately addressed your comments raised in a previous round of review and you feel that this manuscript is now acceptable for publication, you may indicate that here to bypass the “Comments to the Author” section, enter your conflict of interest statement in the “Confidential to Editor” section, and submit your "Accept" recommendation.

Reviewer #1: (No Response)

Reviewer #2: All comments have been addressed

2. Is the manuscript technically sound, and do the data support the conclusions?

Reviewer #1: Yes

Reviewer #2: Yes

3. Has the statistical analysis been performed appropriately and rigorously? 

Reviewer #1: Yes

Reviewer #2: Yes

4. Have the authors made all data underlying the findings in their manuscript fully available?

Reviewer #1: Yes

Reviewer #2: Yes

5. Is the manuscript presented in an intelligible fashion and written in standard English?

Reviewer #1: Yes

Reviewer #2: Yes

6. Review Comments to the Author

Reviewer #1: The authors have addressed all the comments and observations. However, I still have some few observations.

There are still some figures that the texts are not clear. Authors should make these texts legible.

Reviewer #2: The authors have adequately addressed the comments raised in the first round of review. The manuscript is more coherent now. It is interesting research and can be accepted for publication.

7. PLOS authors have the option to publish the peer review history of their article (what does this mean?). If published, this will include your full peer review and any attached files.

Reviewer #1: **Yes: **Mohammed Sanusi Shiru

Reviewer #2: No

---

## [Author Response · Author response to Decision Letter 1]

20 Apr 2021

Author response: Thank you for pointing this out! According to this requirement, we reviewed each reference carefully in the reference list of our manuscript to ensure that it is complete and correct. There are two references with incomplete or incorrect information, which are reference [47] and reference [62], and we have changed them in 'Revised Manuscript with Track Changes' and 'Manuscript'. Then, after detail reviewed, all of our cited paper are not retracted and could check their information in the website of publishers or journals. 

2) Please state in your Methods section whether any permits were obtained for the field work, or if no field permits were obtained, a short statement as to why not.

Author response: The Qinghai-Tibet Plateau has unique topographical landscape and scientific research significance, and the Chinese Academy of Sciences carries out various scientific research investigations here every year. Our Field work obtained the permission from our school and local governments. We have state that permit in “Manuscript” and “Revised Manuscript with Track Changes” from Line200 to Line 204.

3) The authors have addressed all the comments and observations. However, I still have some few observations. There are still some figures that the texts are not clear. Authors should make these texts legible.

Author response: Thank you for point this out, we check all the figures in our manuscript in Electronic and printed document. Then we changed the sizes and colors to ensure their legible in Figure1, Figure2, Figure3 and Figure8. And all the figures will be upload to the website, if you have any further questions, please contact us again.

---

## [Decision Letter · Decision Letter 2]

4 May 2021

Analysis of the spatial association of geographical detector-based landslides and environmental factors in the southeastern Tibetan Plateau, China

PONE-D-20-11323R2

Dear Dr. jia,

We’re pleased to inform you that your manuscript has been judged scientifically suitable for publication and will be formally accepted for publication once it meets all outstanding technical requirements.

Kind regards,

Shamsuddin Shahid

Academic Editor

PLOS ONE

Additional Editor Comments (optional):

Reviewers' comments:

Reviewer's Responses to Questions

**Comments to the Author**

1. If the authors have adequately addressed your comments raised in a previous round of review and you feel that this manuscript is now acceptable for publication, you may indicate that here to bypass the “Comments to the Author” section, enter your conflict of interest statement in the “Confidential to Editor” section, and submit your "Accept" recommendation.

Reviewer #1: All comments have been addressed

2. Is the manuscript technically sound, and do the data support the conclusions?

Reviewer #1: Yes

3. Has the statistical analysis been performed appropriately and rigorously? 

Reviewer #1: Yes

4. Have the authors made all data underlying the findings in their manuscript fully available?

Reviewer #1: (No Response)

5. Is the manuscript presented in an intelligible fashion and written in standard English?

Reviewer #1: Yes

6. Review Comments to the Author

Reviewer #1: Authors have addressed the comments. Figures have been significantly improved and manucript can be considered for publication.

7. PLOS authors have the option to publish the peer review history of their article (what does this mean?). If published, this will include your full peer review and any attached files.

Reviewer #1: **Yes: **Mohammed Sanusi Shiru

---

## [Editor Report · Acceptance letter]

11 May 2021

PONE-D-20-11323R2 

Analysis of the spatial association of geographical detector-based landslides and environmental factors in the southeastern Tibetan Plateau, China 

Dear Dr. Jia:

I'm pleased to inform you that your manuscript has been deemed suitable for publication in PLOS ONE. Congratulations! Your manuscript is now with our production department. 

Kind regards, 

on behalf of

Dr. Shamsuddin Shahid 

Academic Editor

PLOS ONE